# Sequences at gene segment termini inclusive of untranslated regions and partial open reading frames play a critical role in mammalian orthoreovirus S gene packaging

Debarpan Dhar[1,2], Samir Mehanovic[2], Walter Moss[3], Cathy L. Miller [1,2]*

1 Interdepartmental Microbiology Graduate Program, Iowa State University, Ames, Iowa, United States of America, 2 Department of Veterinary Microbiology and Preventive Medicine, College of Veterinary Medicine, Iowa State University, Ames, Iowa, United States of America, 3 Roy J. Carver Department of Biochemistry, Biophysics and Molecular Biology, Iowa State University, Ames, Iowa, United States of America

* clm@iastate.edu

**Data Availability Statement:** The data that support the findings of this study are publicly available from

## Abstract

Mammalian orthoreovirus (MRV) is a prototypic member of the *Spinareoviridae* family and has ten double-stranded RNA segments. One copy of each segment must be faithfully packaged into the mature virion, and prior literature suggests that nucleotides (nts) at the terminal ends of each gene likely facilitate their packaging. However, little is known about the precise packaging sequences required or how the packaging process is coordinated. Using a novel approach, we have determined that 200 nts at each terminus, inclusive of untranslated regions (UTR) and parts of the open reading frame (ORF), are sufficient for packaging S gene segments (S1-S4) individually and together into replicating virus. Further, we mapped the minimal sequences required for packaging the S1 gene segment into a replicating virus to 25 5′ nts and 50 3′ nts. The S1 UTRs, while not sufficient, were necessary for efficient packaging, as mutations of the 5′ or 3′ UTRs led to a complete loss of virus recovery. Using a second novel assay, we determined that 50 5′ nts and 50 3′ nts of S1 are sufficient to package a non-viral gene segment into MRV. The 5′ and 3′ termini of the S1 gene are predicted to form a panhandle structure and specific mutations within the stem of the predicted panhandle region led to a significant decrease in viral recovery. Additionally, mutation of six nts that are conserved across the three major serotypes of MRV that are predicted to form an unpaired loop in the S1 3′ UTR, led to a complete loss of viral recovery. Overall, our data provide strong experimental proof that MRV packaging signals lie at the terminal ends of the S gene segments and offer support that the sequence requirements for efficient packaging of the S1 segment include a predicted panhandle structure and specific sequences within an unpaired loop in the 3′ UTR.

## Author summary

Mammalian orthoreovirus (MRV) faces the daunting challenge of incorporating one copy of ten unique double-stranded RNA segments into the mature virion. Despite substantial

ISU DataShare with the identifier(s) https://doi.org/10.25380/iastate.24052284.v2.

**Funding:** This work was funded by National Institute of Health (NIH, www.nih.gov) R15CA202984 and Iowa State University College of Veterinary Medicine (https://vetmed.iastate.edu/) Seed Grants to CLM. In addition to the grant funding, DD, SM, and CLM recieved salary support from the Iowa State College of Veterinary Medicine. The funders did not play any role in the study design, data collection and analysis, decision to publish, or preparation of the manuscript.

**Competing interests:** The authors have declared no competing interests exist.

prior research, we understand very little about how the selective packaging mechanism in MRV works and have identified few signal sequences within the MRV genome that influence packaging. A significant reason for this lack of understanding is that deletions or mutations of viral proteins and the genome are not well tolerated by the virus. Using novel approaches, we have determined the importance of gene segment terminal ends in packaging four MRV gene segments. Furthermore, we provide new information towards identifying the minimum stretch of nucleotide sequences sufficient for packaging one of the segments. Additionally, for the first time, we lend experimental support to the significance of a predicted RNA secondary structure and conserved sequences within that structure in the packaging process. With this work, we expect to fill knowledge gaps in our understanding of MRV genome packaging. With the developing potential of MRV as an oncotherapeutic vector, this information is expected to inform the rational design of MRV as a gene delivery vehicle.

## Introduction

The process of gene segment packaging, whereby one copy of each of multiple gene segments are faithfully incorporated into a full genome, a virion is assembled, and the genome is replicated remains a persistent mystery in our understanding of the basic biology of segmented double-stranded (ds) RNA viruses in the Order *Reovirales*. Mammalian orthoreovirus (MRV) has long been studied as a prototypical member of this Order which also includes important animal and human pathogens such as bluetongue virus (BTV) and rotavirus (RV). MRV is considered clinically benign, however it is a potent oncolytic virus that has been widely studied as a cancer therapeutic in Phase I-III clinical trials and has recently been awarded FDA fast track designation for breast and pancreatic cancer therapy [1–3]. Therefore, it is critical to resolve remaining questions in the MRV life cycle to both answer longstanding important biological questions and provide actionable insight into its development as a cancer therapeutic.

MRV has ten dsRNA genome segments [three large (L), three medium (M), and four small (S)] that encode for twelve proteins. Eight of the twelve proteins form the bilayered virus capsid, within which the ten dsRNA genome segments are packaged [4]. During genome packaging, viruses with multi-segmented genomes must coordinate the faithful incorporation of one copy of each gene segment into the virion. This may be achieved through two fundamental strategies, one where the gene segments are "fed" into a preassembled empty viral capsid and another where the gene segments contribute to a scaffold around which viral proteins assemble [5–7]. For φ6 bacteriophage of the *Cystoviridae* family that have a genome of three dsRNA segments, each genome segment is fed in a specific order (Small (s+)>Medium (m+)>Large (l+)) into an assembled procapsid by a helicase-dependent mechanism. The packaging process is dually dependent on a conserved 18 nt sequence stretch present in all three segments and unique packaging (pac) sequences located at the 5′ end of each of the segments [8–10]. During φ6 packaging, the s+ gene pac sequence binds to a specific site on the procapsid vertices where it is then translocated into the procapsid. Packaging of the s+ segment results in conformational changes and expansion of the procapsid that reveal recognition sites necessary for packaging of the m+ segment, and following additional procapsid expansion, the l+ segment.

For such a mechanism to work in dsRNA viruses with larger genomes (9-12 segments), it would necessitate up to 12 conformational changes of the preformed core to package the RNA [6]. As empty core particles are structurally indistinct from those containing a full genome, it is unlikely that members of the *Spinareoviridae* or *Sedoreoviridae* families use such a

mechanism for packaging. Instead, evidence suggests that for segmented dsRNA viruses with larger genomes, packaging is likely guided through RNA:RNA interactions between the plus-strand RNAs of each segment. For example, *in vitro* studies revealed that RNA interactions between the ten plus-strand RNA segments of BTV are essential for assembling the viral core [11], and genome packaging is a sequential process, likely triggered by the UTR of the smallest gene segment (S10) that subsequently recruits the medium and larger segments via RNA:RNA interactions to form an assortment complex [12]. For RV, capsid proteins assemble around a plus-strand RNA/capping enzyme/RNA-dependent RNA Polymerase (RdRP) complex. This assembly is likely enhanced and stabilized by the interactions of the capsid protein N terminal extensions with the +ssRNAs [13]. RV plus-strand RNA segments also have been shown to interact with each other by RNA:RNA interactions. Packaging likely follows a sequential order with the smaller S segments initiating the process. In addition, there is a reduction in viral complex formation and replication when the UTRs are targeted using sequence-specific anti-sense oligoribonucleotides (ORNs) [14].

Another line of evidence that suggests that MRV uses a selective packaging mechanism is that the ratio of total particles to infectious particles for MRV is low [15–18], indicating that the MRV packaging mechanism efficiently produces progeny virions with an entire complement of genes. In addition, structural studies on MRV and other dsRNA viruses have shown that the number of transcription complexes usually correlates with the number of gene segments, despite structural space for additional complexes [19–22]. Altogether, these data lend support that MRV uses a selective genome packaging mechanism for producing progeny virions.

The selective packaging mechanism of MRV can be imagined as a process made up of three fundamental processes. First is *Assortment*, where the individual plus-strand RNAs for all ten genome segments (S1-S4, M1-M3, and L1-L3) interact to form a sorted set of the full genome. *Assembly* of the inner capsid proteins to form the viral core either quickly follows or coincides with assortment. Recent evidence from MRV serotype 3 and RV suggests that a single-layered particle (SLP) may form as an assembly intermediate, and that the assortment complex is brought to the SLP by the RdRp (λ3, MRV; VP1, RV), which is bound to each gene segment terminus [23,24]. Second strand/negative-strand synthesis (*Replication*) of the plus-strand RNAs to dsRNAs by λ3 and a polymerase co-factor µ2 is thought to occur during or shortly after viral core assembly [4].

MRV genomes have short UTRs that range from 12-32 nts (5′ end), and 35-83 nts (3′ end), with a string of 4 (5′ GCUA) and 5 (3′ UCAUC) nts at the termini of each segment that are conserved in all serotypes [25–28]. Similar terminal short conserved sequences are critical for RV polymerase binding, suggesting their primary function is in replication, and not assortment/assembly [29]. Moreover, defective interfering particle gene segments with large internal gene deletions always retain the UTRs and variable lengths of the ORF at each end [28,30], suggesting that packaging signals extend beyond the UTRs into the ORF. Based on published data from MRV and other viruses with a dsRNA genome [28,30–35], we have hypothesized that signal sequences present at the terminal ends of each MRV gene segment inclusive of UTRs and partial ORFs guide the overall process of packaging (assortment, assembly, and replication).

As a result of the extension of packaging signals into the ORF, and the dependence of successful MRV progeny production on the expression of all viral proteins, identifying packaging signals experimentally in the context of independently replicating virus has proven challenging. Initial experiments to identify MRV packaging sequences relied on a complicated reverse genetics (RG) system in which a plasmid encoding the chloramphenicol acetyltransferase (CAT) gene was flanked on each end with MRV gene segment terminal sequences, followed by recombinant virus rescue using a helper virus and cell lines expressing the protein encoded

by the deleted ORF [31–34]. More recently, a novel approach where MRV gene segment protein-coding regions were unlinked from putative packaging signals by duplicating terminal regions inclusive of the ORF and UTR, then introducing mutations in the third/wobble codon position in the ORF region was used successfully to identify sequences involved in MRV packaging [35]. The prior research was rigorous and generated significant novel information highlighting the importance of the terminal regions of the MRV genes towards packaging. For example, these experiments indicated that the terminal 200 nts in S2, S4, M1 and L1 genes of the Type 3 Dearing serotype likely play a role in packaging. However, these approaches were experimentally complex, relying on helper cell lines or the addition of duplicated sequences. Furthermore, in some cases these approaches could not rule out the possibility of wildtype (WT) sequences within the internal regions of the ORF contributing to the packaging process. Moreover, these approaches often did not identify minimal RNA sequences or structures required for the packaging process.

In this work we have created a novel approach, termed Wobble/Block Replacement (W/BR), that uses codon redundancy to introduce the maximum possible nt changes across the ORF of each MRV gene segment without altering amino acid sequences. Using this approach, we have identified the importance of terminal nucleotide (nt) sequences in the packaging of S gene segments (S1-S4) of the Type 1 Lang (T1L) serotype. Furthermore, for the first time, we have identified minimal sequences in the T1L S1 gene that are sufficient for packaging. We have also provided experimental support that sequences within a predicted panhandle structure formed by the interaction of the terminal ends of the MRV gene play an important role in S1 packaging. Finally, we have used a novel Segment Incorporation Assay to investigate the sufficiency of packaging signal sequences in incorporating a non-viral reporter gene into infectious MRV. Our results are interesting from a fundamental biology standpoint in shedding light on the black box of dsRNA virus packaging. Additionally, because MRV is being investigated as an oncolytic virus, our findings may be of use clinically by identifying packaging sequences that are indispensable for the creation of MRV oncotherapeutic vectors.

## Results

### Wobble/Block Replacement (W/BR) is a novel approach to identify MRV packaging sequences

A central problem in determining packaging sequences within the MRV genome is that predicted sequences necessary for assortment/assembly/replication extend beyond the UTRs and into the protein-coding region (Fig 1A, Top, PAC). It is difficult to introduce mutations or deletions into gene segments without impacting MRV proteins, making the identification of sequences important for packaging in the context of replication-competent virus challenging. Our novel W/BR approach overcomes these limitations by taking advantage of codon redundancy to make maximum nt changes throughout the ORF (Fig 1A, Bottom) while ensuring that the amino acid sequence remains unchanged. Using this approach, we designed S1-S4 mutant constructs (18/S1/40, 23/S2/59, 32/S3/73, 37/S4/69) where each gene contained WT sequence in both 5′ and 3′ UTRs as well as start and stop codons, but every other possible nt within the ORF was "wobbled". These "wobble" changes were not limited to nts in the third/wobble position but include double nt changes in leucine and arginine and triple nt changes in serine codons. Wobble mutant constructs have been named uniformly throughout the publication in the format X/Gene/Y, where X and Y denote the number of WT sequences at the 5′ and 3′ end, respectively, of a wobbled gene segment. As a result of the wobble changes within the ORF, the four S gene segments are only 57 to 61% identical to their corresponding WT gene sequences (Fig 1A, bottom). Plasmids are also made in which terminal regions of varying

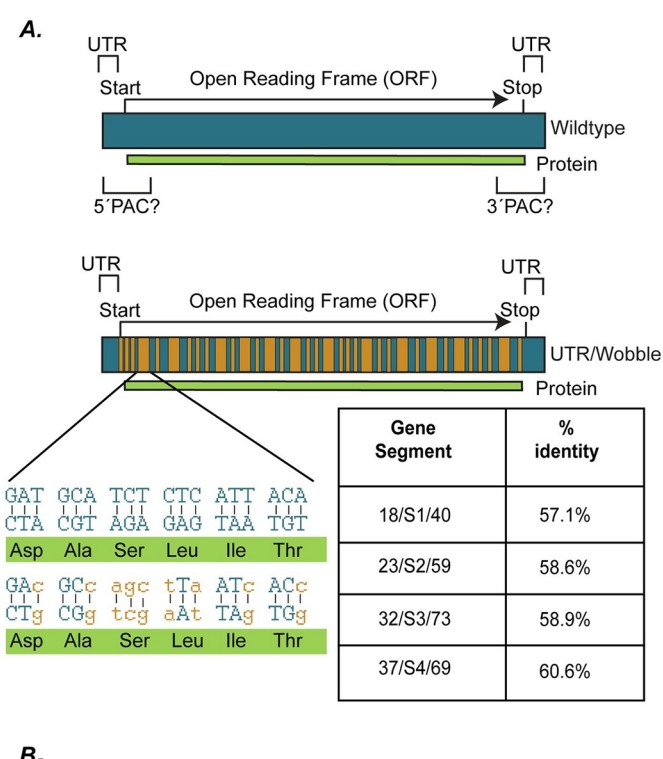

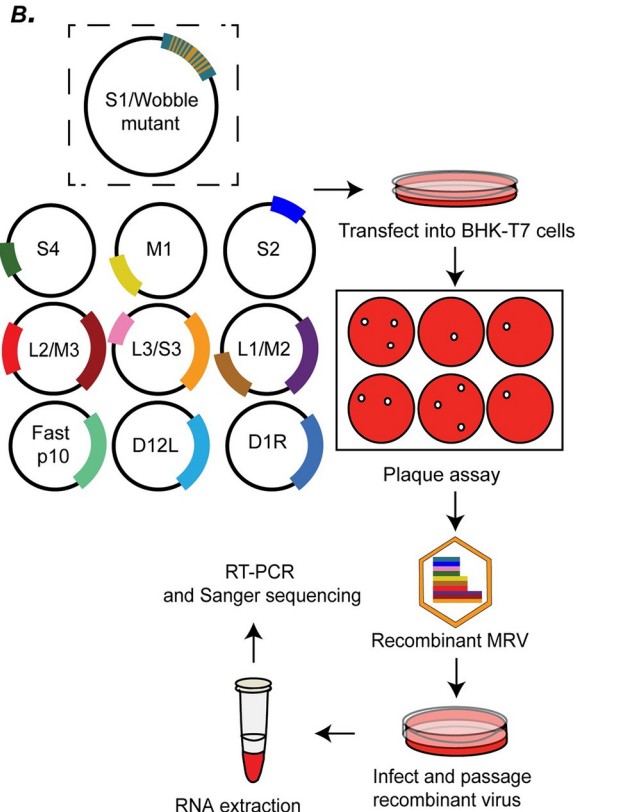

**Fig 1. Wobble/Block replacement is a novel approach to identify MRV packaging sequences.** (A) (Top) Illustration of an MRV WT S gene segment showing short UTRs and predicted packaging signals that extend beyond the UTR into the open reading frame. (Bottom) Illustration of wobble mutation strategy with WT nt regions represented in green, and wobble mutations represented in orange. An example of wobble mutations and the percent change to the nt sequences compared to WT are shown. (B) Schematic of the W/BR assay showing replacement of WT S1 with an S1

wobble mutant. The S1 wobble mutant is transfected with plasmids encoding for the remaining nine WT MRV gene segments and accessory plasmids into BHK-T7 cells in the MRV RG assay. Viral titer is determined using plaque assay, and isolated plaques are picked and passaged to amplify the viral titer before RNA extraction, reverse transcription, PCR amplification, and sequencing are completed.

lengths within the ORF maintain WT sequences to identify packaging requirements at the sequence level. To determine packaging requirements, each of these constructs are subjected to the MRV reverse genetics (RG) assay and the impact of the mutations on recombinant virus recovery is determined by measuring viral titer using standard MRV plaque assays. Mutant plasmids expressing gene segments that contain nt sequences sufficient for packaging generate replicating viruses that form plaques on L929 cells, which are picked and passaged for amplification, viral RNA extraction, reverse transcription polymerase chain reaction (RT-PCR), and sequencing (Fig 1B).

## 200 WT nts but not UTRs at each terminus are sufficient for efficient recovery of replicating virus

To provide proof of concept of the W/BR assay, sets of two wobble mutant constructs were designed for each S gene segment, S1 (encoding σ1 and σ1s), S2 (encoding σ2), S3 (encoding σNS) and S4 (encoding σ3). In each set, one construct maintained WT sequences only at the terminal ends comprising the UTR, Start codon and 2 non-wobbled nts (5′ end) and UTR and Stop codons (3′ end) while the ORF was fully wobbled (18/S1/40, 23/S2/59, 32/S3/73, 37/S4/69). In the second construct in each set, 200 nt sequences at each terminus, inclusive of the UTRs, Stop and Start codons, and variable lengths of the ORF maintained WT nt sequences, while the rest of the ORF remained fully wobbled (200/S1/200, 200/S2/200, 200/S3/200, 200/S4/200) (Fig 2A). Each set of mutant clones for S1, S2, S3, and S4 genes and their WT gene counterparts were subjected to RG assays, and the viral titer of each RG lysate was determined by plaque assay on L929 cells. No plaques were formed following RG with 23/S2/59, 32/S3/73, or 37/S4/69 constructs, strongly suggesting the UTRs are insufficient to support recovery of viruses containing wobble mutations across the entire ORF (Fig 2B). For the S1 gene, the 18/S1/40 virus was recovered at levels 10,000-100,000 fold lower than the WT S1, and produced significantly smaller plaques than WT (Fig 2C), suggesting that while virus recovery was not completely lost when mutations were introduced across the ORF of this gene segment, it was severely and significantly decreased. On the other hand, using plasmid constructs in the RG assay where 200 nt at each terminus were WT with maintenance of internal "wobble" sequences (200/S1/200, 200/S2/200, 200/S3/200, and 200/S4/200) resulted in recombinant virus production at levels not significantly different from the corresponding WT plasmids in each case (Fig 2B). These results indicate that when the ORF nt sequence is mutated substantially by wobble introduction, the UTRs are unable to support efficient rescue of viruses with mutations across the ORF of individual S genes. However, even though wobble mutations are maintained across the majority of the gene segment, 200 WT nts at each terminal end, inclusive of the UTRs and parts of the ORF, are sufficient for the recovery of replicating viruses to WT levels.

## UTR-only wobble mutants are defective in a late step in the virus replication cycle

Though the WT amino acid sequence of each gene was maintained upon the introduction of wobble mutations, there is a possibility that defects in protein expression of the σ proteins were altered in our UTR mutant clones, which could impact virus rescue. To examine this

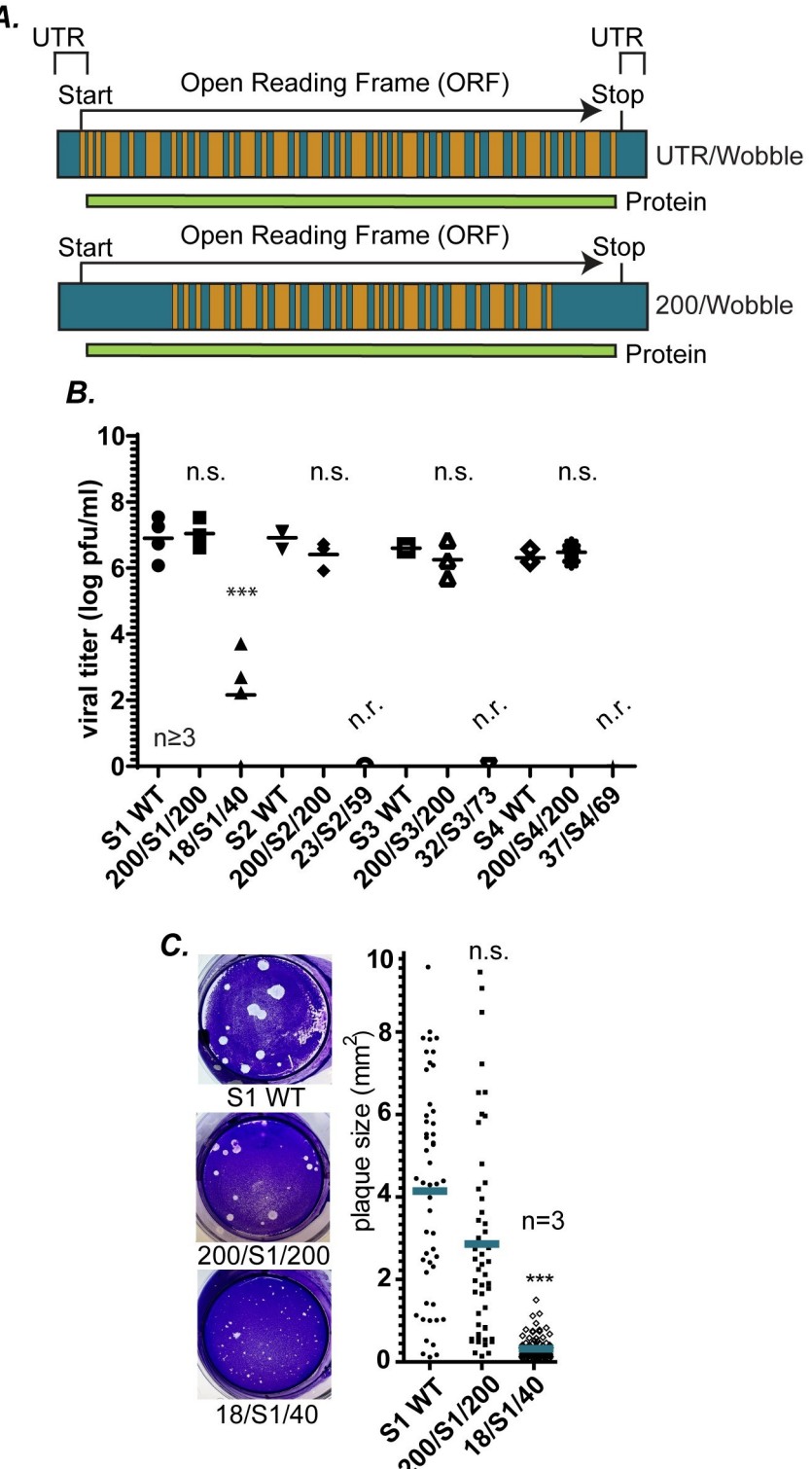

**Fig 2. 200 WT nts but not WT UTRs at each terminus are sufficient for efficient recovery of replicating virus.** (A) Diagram showing constructs that maintain WT UTRs but are fully "wobbled" across the remaining gene (UTR/Wobble) and constructs containing 200 WT sequences at both the terminal ends of the S gene segments (200/Wobble) for each T1L MRV S gene segment. (B) Viral titers [log plaque forming units (pfu)/ml] of the S WT, 200/S gene/200, and UTR/S gene/UTR gene segments (S1-S4) measured after RG and plaque assay. (C) Viral plaque size images (left) and quantification (right) of WT S1, 200/S1/200, and 18/S1/40 recombinant viruses recovered from RG. (n=independent biological replicates, n.r.=virus not recovered, n.s.=non-significant, ***P<0.001).

possibility, S gene segment wobble mutant plasmids were transfected into BHK-T7 cells, and at 24 hours post-transfection, cells were processed for immunoblotting using protein-specific antibodies to measure protein expression. Despite vastly different efficiencies in virus recovery, there were no significant differences in σ1 protein expression in pT7 S1 WT, pT7 200/S1/200 and pT7 18/S1/40 (Fig 3A). When measuring σ2 expression, we noted a significant decrease in expression of the pT7 23/S2/59 construct relative to pT7 200/S2/200, although there was no significant difference from WT pT7 S2 (Fig 3B). To determine whether a decrease in σ2 protein expression contributed to the loss of virus recovery of 23/S2/59 virus (Fig 3B), we performed a transcomplementation assay. In this assay, RG was performed with the addition of a plasmid expressing the WT σ2 protein from a construct that lacks the T7 promoter and Hepatitis δ ribozyme. Therefore, the RNA produced does not resemble the authentic S2 gene segment and cannot be incorporated in place of S2 into the MRV genome for virus rescue. The addition of this plasmid in the RG assay resulted in a similar expression of σ2 protein across all samples (Fig 3E) but did not result in 23/S2/59 virus recovery (Fig 3F). This suggests that increasing σ2 protein expression in RG assays using pT7 23/S2/59 does not result in the recovery of the virus and that the defect in recovery lies outside of protein expression. A second possibility we considered for lack of RG recovery of 23/S2/59 virus was RNA expression. To determine whether the defect in viral recovery of the 23/S2/59 virus was due to a lack of S2 RNA available for packaging, we performed an RT-qPCR assay to measure the relative RNA levels of S2 (Fig 3G). The accumulated RNA from the pT7 23/S2/59 and pT7 200/S2/200 construct after 24 hours of transfection was normalized to the housekeeping gene β-actin*(ACTB)*. The normalized 23/S2/59 RNA was compared to the reference construct of normalized 200/S2/200 RNA. Our results showed no significant changes in the RNA levels of 23/S2/59 compared to 200/S2/200.

While 200/S3/200 virus was recovered to WT levels, the 32/S3/73 virus was not recovered (Fig 2B), however, pT7 200/S3/200 and pT7 32/S3/73 express σNS protein similarly with respect to each other (Fig 3C), suggesting that protein expression levels are not playing a role in the loss of 32/S3/73 recovery. However, it is interesting to note that while 200/S3/200 and S3 WT viruses were recovered from RG at similar levels (Fig 2B), the σNS protein was expressed at significantly higher levels from pT7 S3 relative to either pT7 200/S3/200 or pT7 32/S3/73 (Fig 3C). This has little bearing on our results but suggests that pT7 S3 expresses σNS at much higher levels than are needed for efficient virus recovery. Similarly, we did not see significant differences in protein expression in immunoblots of σ3 between 200/S4/200 and 37/S4/69 (Fig 3D). In the above experiments, the mutant plasmids were expressed alone in transfected cells. To rule out the possibility that another viral protein may be altering expression of the S genes in our RG assays, we additionally performed protein expression assays on a subset of our WT and mutant plasmids (S1 and S3) in cells transfected with all the viral plasmids used in RG assays. At 24 h post-transfection, cells were processed and subjected to immunoblotting using protein-specific antibodies. The relative expression levels of mutant and WT σ1 and σNS were similar to that measured when these plasmids were expressed in the absence of other viral proteins (compare Fig 3H to Fig 3A and 3C). Taken altogether, this data strongly suggests changes in protein expression are not responsible for the losses in virus rescue of the S gene UTR-only mutants in RG assays (Fig 2B).

While the data presented in Fig 3A–3H suggests there is no defect in transcription or translation from the plasmid constructs used in RG assays, it remained possible that the UTR-only mutants can form genome-full virions that are unable to initiate infection and are therefore undetectable in plaque assays. While we were unable to recover 23/S2/59, 32/S3/73 or 27/S4/69 viruses to levels detectable in plaque assays, we could recover low levels of 18/S1/40 virus. The small plaque phenotype (Fig 2C) of this virus illustrates an evident defect in the virus life cycle,

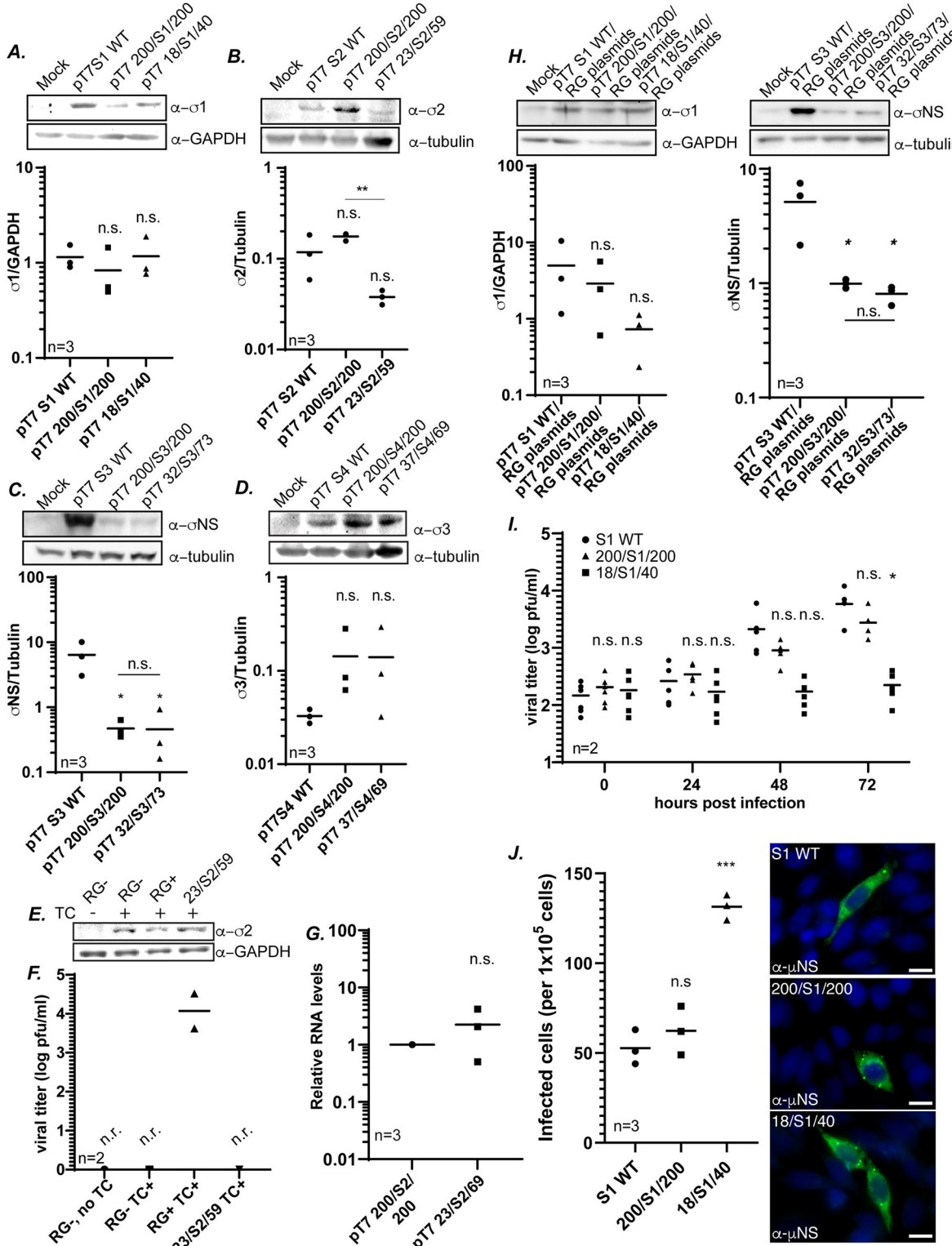

**Fig 3. UTR-only wobble mutants are defective in a late step in the virus replication cycle** Representative immunoblot (top) and corresponding quantification (bottom) of σ protein expression normalized to loading controls 24 hours post transfection of (A) pT7 S1 WT, pT7 200/S1/200 and pT7 18/S1/40 (B) pT7 S2 WT, pT7 200/S2/200 and pT7 23/S1/59 (C) pT7 S3 WT, pT7 200/S3/200 and pT7 32/S1/73, and (D) pT7 S4 WT, pT7 200/S4/200 and pT7 37/S4/69. (E) Representative immunoblot of σ2 expression from transcomplementation RG lysates. (F) Viral titer (log pfu/ml) from RG lysates measured by plaque assay in the σ2 transcomplementation

assay (G) Relative S2 RNA levels measured using RT-qPCR and normalized to housekeeping gene *β-actin* (*ACTB*). (H) Representative immunoblot and corresponding quantification of σ1 and σNS expression normalized to loading controls 24 hours post co-transfection of all RG plasmids. (I) Viral titer (log pfu/ml) measured at 0, 24, 48 and 72 hours p.i. of S1 WT, 200/S1/200 and 18/S1/40 at an MOI of 0.001 in L929 cells (J) Number of infected cells at 24 h p.i. (left) and representative immunofluorescence assay images (right) from L929 cells infected with S1 WT, 200/S1/200 and 18/S1/40 viruses at an MOI of 0.001 and stained with μNS antisera (green). DAPI (blue) is also shown. (Scale bar 10μm). TC: Transcomplementation, RG-: Reverse genetics negative control (no pT7 S2). RG+: Reverse genetics positive control (pT7 S2 WT). One-way ANOVA and Tukey comparisons were used for statistical analyses of data, except for Fig 3I where Two-way ANOVA and Tukey multiple comparisons was performed. n=independent biological replicates, n.r.=virus not recovered, *$p<0.05$, ** $p<0.01$, *** $p<0.001$, n.s.=non-significant, $p>0.05$.

and accordingly, attempts at amplifying 18/S1/40 to high titers were unsuccessful. However, we were able to generate limited amounts of virus, which allowed us to perform experiments to further investigate the stage in the virus life cycle that is impaired in this mutant. First, we performed a multiple-cycle replication assay comparing growth over time of S1 WT, 200/S1/200, and 18/S1/40. L929 cells were infected with each virus using the same starting titer. An MOI of 0.001 was used because of the low titer of the 18/S1/40 stocks we were able to generate. Samples were taken at 0, 24, 48, and 72 hours and subjected to plaque assay on L929 cells to measure virus growth over time. At the 0 timepoint, 18/S1/40 formed similar numbers of much smaller plaques relative to WT and 200/S1/200, indicating it can enter cells and initiate infection normally. However, 18/S1/40 virus replicated very slowly relative to WT and 200/S1/200 over time, with the difference in viral titers between 18/S1/40 and S1 WT being significant by 72 hours post infection (Fig 3I). This suggests that 18/S1/40 virions can enter cells and initiate infection but are defective at a later step in the replication cycle. To assess virus entry, transcription, and translation more directly, we also performed immunofluorescence assays comparing the number of infected cells in 18/S1/40 relative to S1 WT and 200/S1/200. L929 cells were infected at an MOI of 0.001, and at 24 h p.i., cells were fixed and stained with antibodies against the non-structural μNS protein. Infected cells were counted in each sample, and somewhat surprisingly, there were approximately 3 times more cells infected with 18/S1/40 relative to S1 WT and 200/S1/200 (Fig 3J). This suggests that 18/S1/40 virus is not defective in any early step (entry/transcription/translation) of infection. It also suggests that 18/S1/40 titers are being slightly undercounted in plaque assays, likely because of the small plaque phenotype. This undercounting does not significantly impact our RG assay results, as the differences between S1 WT and 18/S1/40 in those assays is 10,000-100,000-fold. Importantly, the μNS staining levels in the infected cells in this assay appeared very similar (Fig 3J), providing more evidence that there is no defect in 18/S1/40 virus entry, transcription, and translation relative to WT and 200/S1/200. Taken altogether, these findings support that the mutations introduced in the UTR-only mutants in the W/BR assay are impacting a late step in the MRV life cycle within the packaging process of gene segment assortment, core assembly, or replication.

## 200 terminal WT nts are sufficient for packaging all S gene segments into the same replicating virus

In our previous results, 200 WT nts at each terminus were sufficient to package each individual S gene segment in the context of all the other WT gene segments. As we do not understand the extent of flexibility within the packaging process, it remained possible that WT sequences within the other S gene segments were able to use an alternative mechanism to enable the inclusion of the mutated segment in the recombinant viruses. To examine whether 200 nt at each terminus in all four S gene segments could facilitate packaging of all the mutated segments into the same virus, we performed RG using pT7 200/S1/200, pT7 200/S2/200, pT7 200/S3/200, and pT7 200/S4/200 plasmids combined with plasmids expressing WT M and L gene

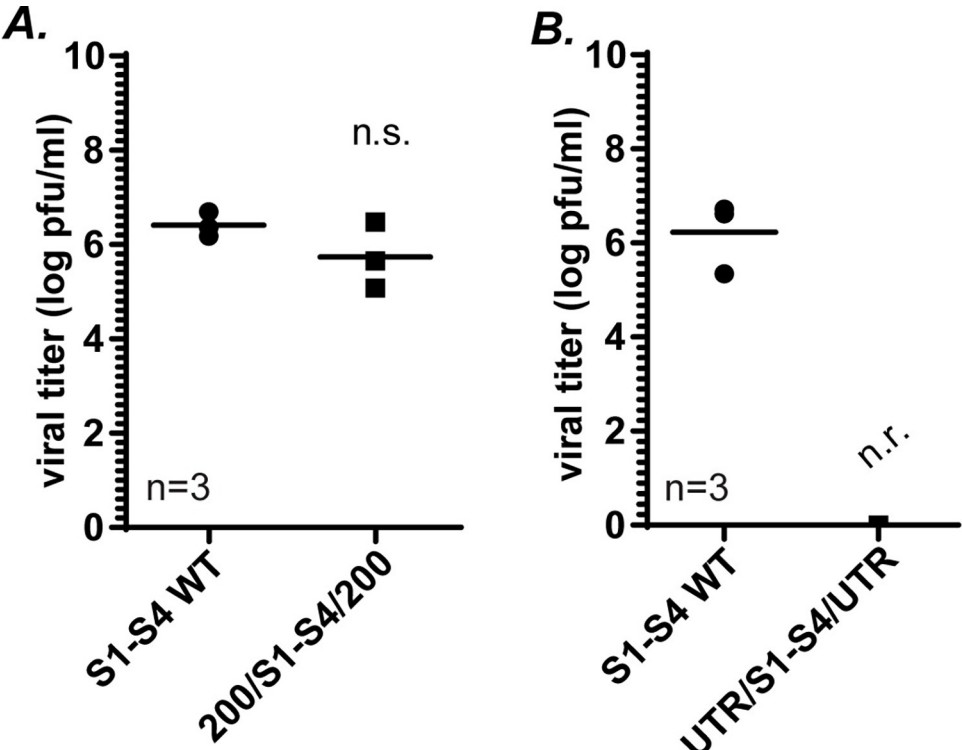

**Fig 4. 200 terminal WT nts are sufficient for packaging all S gene segments into the same replicating virus** Viral titers (log pfu/ml) of recovered recombinant viruses following RG and plaque assay using WT (S1-S4 WT) and (A) 200/S gene/200 S1,S2,S3 and S4 plasmids (200/S1-S4/200) or (B) UTR/S gene/UTR S1,S2,S3 and S4 plasmids (UTR/S1-S4/UTR) with the 6 other WT (M and L genes) plasmids. One-way ANOVA and Tukey comparisons were used for statistical analyses of data. n=independent biological replicates, n.r.=virus not recovered, n.s.=non-significant, p>0.05.

segments and measured the viral titer of the RG lysates using plaque assay. Interestingly, recombinant viruses were recovered at levels not significantly different from samples where all WT plasmids were used (Fig 4A). Similar to results in Fig 2A, when RG was performed using all four S segments possessing wobble mutations throughout the ORF, with only WT UTRs and Stop/Start codons at each terminal end (18/S1/40, 23/S2/59, 32/S3/73, 37/S4/69) no virus was recovered (Fig 4B). Taken together with our previous results, these findings strongly suggest that the signals required for packaging the S gene segments into a replicating virus are localized within the terminal 200 nts of each of the four S gene segments, and that favorable RNA:RNA interactions that occur during packaging between the S genes and other segments, as well as important RNA:protein interactions likely occur within these 200 terminal nt regions.

## 25 5′ and 50 3′ nts are necessary for efficient S1 packaging in the W/BR assay

Our previous results identified a stretch of 200 nts at each terminus sufficient for packaging each of the S gene segments into replicating viruses. However, these nts may not be the minimal number sufficient for packaging. Our long-term goal is to better understand the MRV packaging process by identifying minimum sequences necessary and/or sufficient for packaging each individual MRV gene segment. Towards this end, we further examined the S1 gene by creating a series of plasmids where the terminal WT sequences were progressively narrowed

at both termini from 200 nts to 50 nts (pT7 200/S1/200, pT7 100/S1/100, pT7 50/S1/50) (Fig 5A). These plasmids were subjected to RG assay and cell lysates were used to determine virus recovery by plaque assay. These experiments demonstrated that the 200 WT nts at the terminal ends could be narrowed to just 50 nts of WT sequence without any loss to rescue efficiency, suggesting that in the context of the W/BR assay, the 5′ UTR+37 nt and the 3′ UTR+13 nt are sufficient for packaging the S1 gene into a replicating virus (Fig 5B).

We were curious to determine the individual role of each terminal end in guiding the packaging process. We therefore created additional plasmids containing WT sequences of varying lengths at either the 5′ end (pT7 100/S1/40, pT7 50/S1/40, pT7 25/S1/40) or the 3′ end (pT7 18/S1/100, pT7 18/S1/50) with only WT UTR/Start/Stop sequence at the opposite end of the gene (Fig 5A). These plasmids were subjected to RG assay and cell lysates collected and used in plaque assays to determine the impact of each mutation on virus rescue. From these experiments, we determined that with only 25 WT nts at the 5′ end (25/S1/40), there was no significant difference in the ability to recover virus compared to the fully WT S1. As demonstrated in Fig 2, and repeated in these experiments, when a plasmid with WT sequence at the 5′ end is limited to just the UTR+Start+2 nt (18/S1/40), there is a precipitous drop of almost 4 logs in viral recovery (Fig 5B). These findings indicate that 25 nts at the 5′ end is sufficient for packaging the S1 gene segment, and further, that sequences between nts 18 to 25 play a critical role in the packaging process. When the 3′ end contained varying stretches of WT sequences and the 5′ end was wobbled beyond the UTR/Start+2 nts (18/S1/100, 18/S1/50), there was a significant drop in viral titer of 2-3 log fold compared to WT S1 gene segment (Fig 5B). This again reflects the relative significance of the 5′ end in the packaging process. However, the 3′ end also clearly contributes to S1 gene segment packaging, as recovery of replicating virus dropped by an additional two logs in the 18/S1/40 construct, which differs by only 10 nts from 18/S1/50. Overall, our data indicate that in the context of our W/BR assay, 25 nts at the 5′ end and 50 nts at the 3′ end are sufficient for efficient packaging of the S1 gene.

To determine if differences in viral rescue are due to differences in σ1 protein expression, each of the S1 mutant plasmids were transfected into BHK-T7 cells followed by immunoblotting using antibodies against σ1. These experiments revealed no significant differences in σ1 expression (Fig 5C and 5D) suggesting that differences in viral titer cannot be attributed to any difference in the translation of the S1 wobble mutant constructs. Additionally, to rule out the possibility that differences in viral recovery are due to differences in RNA availability, we measured relative RNA levels using RT-qPCR. BHK-T7 cells were transfected with the S1 mutant plasmids and 24 hours post-transfection, cells were collected and the accumulated levels of S1 RNA were measured and normalized to the housekeeping gene *β-actin* (*ACTB*). The normalized RNA levels of the S1 mutant constructs were made relative to the reference gene of 200/S1/200 for which viruses were recovered to WT S1 levels (Fig 2B). Similar to the protein expression data, RT-qPCR showed no significant differences in relative RNA levels of the mutant S1 constructs (Fig 5E). This suggests that differences measured in viral recovery are likely due to the effect of mutations on the process of packaging (assortment, assembly and/or replication), and not due to differences in transcription or translation of the mutant constructs.

## 5′ and 3′ UTRs are necessary for S1 gene packaging

In our previous results (Fig 2B), we found that there is minimal rescue of recombinant virus when the S1 gene contains a wobbled ORF and WT UTRs (18/S1/40), suggesting that the UTRs are not sufficient for optimal packaging of the S1 gene. However, these experiments do not rule out the likely possibility that the UTRs play a significant role in the packaging process. To specifically address this question, we created plasmids where UTR purine nts were mutated to

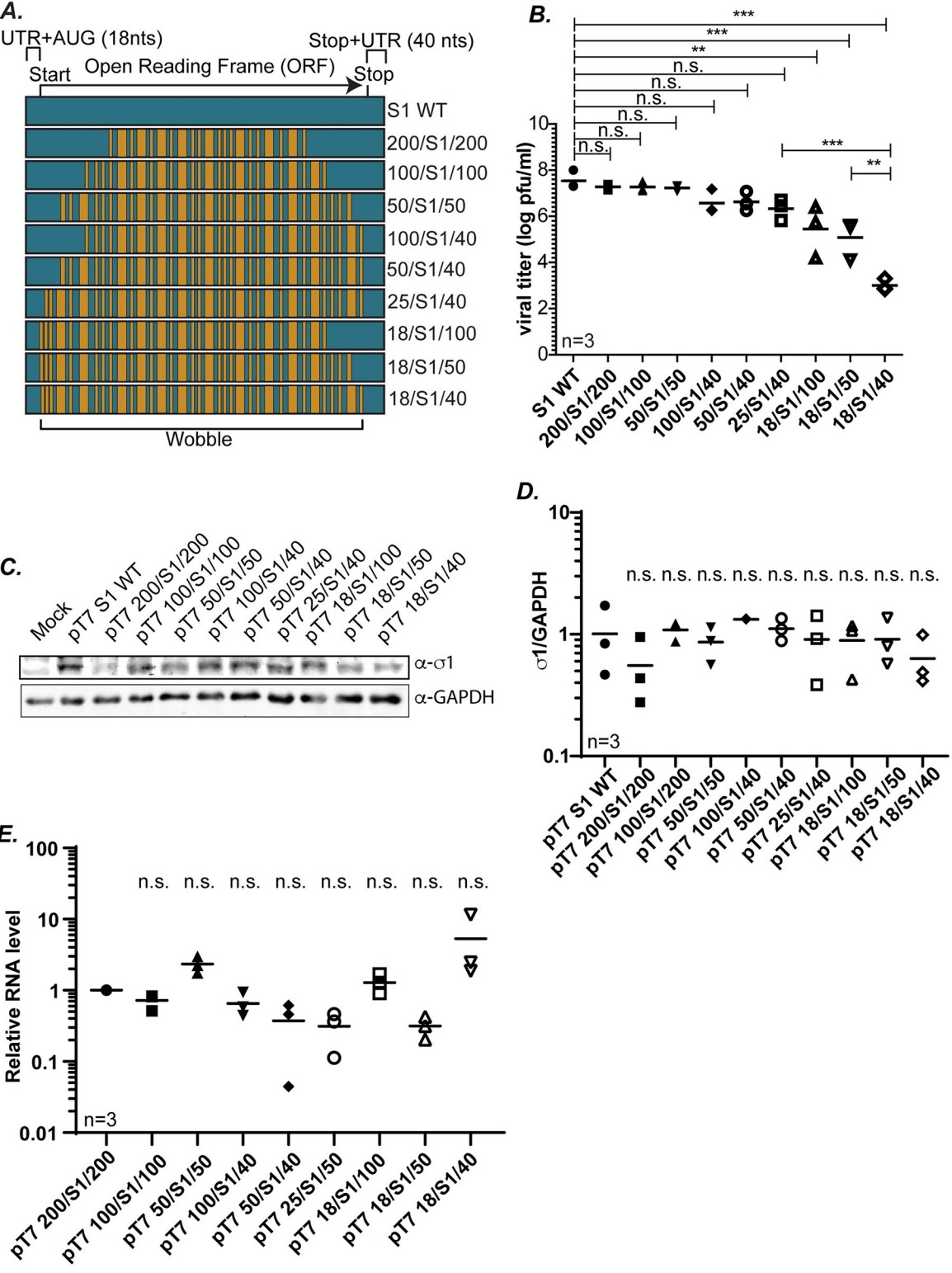

**Fig 5. 25 5′ and 50 3′ nts are necessary for efficient S1 packaging in the W/BR assay** (A) Illustration of S1 gene segment mutant constructs designed to determine the minimum sequences required for packaging of the S1 gene. (B) Viral titers (log pfu/ml) determined from RG followed by plaque assay for indicated S1 mutant gene segments. (C) Representative immunoblots of cells transfected with indicated S1 gene plasmids at 24 h post-transfection using σ1 and GAPDH antibodies. (D) Quantification of biological replicates of immunoblots in (C). σ1 protein intensities are normalized to the corresponding GAPDH loading controls. (E) Relative S1 RNA levels measured using RT-qPCR and normalized to housekeeping gene β-actin. All statistical analysis were done using one way ANOVA and Tukey multiple comparison. **p<0.01, ***p<0.001, n.s.=non-significant, p>0.05.

pyrimidines and vice versa at either the 5′ UTR or 3′ UTR, or at both the 5′ and 3′ UTRs (Fig 6A). In all the UTR mutant constructs, the 5′ GCUA and 3′ UCAUC conserved sequences that are predicted to bind the polymerase complex to guide negative-strand synthesis remained WT [29,36]. Each of these plasmids was used as a replacement for pT7 S1 WT in RG assays, and viral titer was determined on the cell lysates by plaque assay. Regardless of whether mutations were present within the 5′ UTR, 3′ UTR or both the UTRs, no recombinant viruses were recovered (Fig 6B).

To rule out differences in protein expression as a factor in loss of recombinant virus rescue, pT7 5′ UTR mut, pT7 3′ UTR mut, and pT7 5′ 3′ UTR mut plasmids were transfected into BHK-T7 cells, followed by immunoblotting using antibodies against σ1. There were no significant differences in protein expression between these constructs and pT7 S1 WT, suggesting the loss in virus rescue was not a result of decreased protein expression (Fig 6C and 6D). Furthermore, there were no significant differences in gene expression of S1 from pT7 5′ UTR mut, pT7 3′ UTR mut, and pT7 5′ 3′ UTR mut as compared to S1 WT as measured by RT-qPCR, suggesting that the lack of virus recovery cannot be attributed to differences in S1 RNA levels between the mutant and WT plasmid constructs (Fig 6E). Taken together with our earlier findings, these results suggest that while the UTRs are not sufficient to support the packaging process of the S1 gene segment (Figs 2B and 5B), they are necessary for this process as mutations within the S1 UTRs at either terminus leads to a total loss of virus recovery.

## 50 WT 5′ and 3′ terminal nts of the S1 gene are sufficient to package a non-MRV gene into infectious virions

Our previous results indicate that 25 nts at the 5′ end and 50 nts at the 3′end are sufficient for packaging the S1 gene segment into replicating viruses in the W/BR assays (Figs 2B and 5B). However, the possibility remains that WT nt sequences remaining within the ORF after wobble mutation introduction continue to contribute to packaging in the W/BR assay. To determine if remaining internal WT sequences play a role in the packaging process, we developed a novel Segment Incorporation Assay. For this assay, plasmids were created in which the S1 ORF was replaced with the ORF for NanoLuc (NL) luciferase flanked on each end with putative S1 packaging sequences identified from our previous results. Six plasmids, which contained the NL ORF flanked on each end with S1 5′ and 3′ terminal sequences were created (pT7 S1 200/NL/200, pT7 S1 100/NL/100, pT7 S1 50/NL/50, pT7 S1 25/NL/50, pT7 S1 25/NL/40, or pT7 S1 18/NL/40). These chimeric S1/NL plasmids were used to replace pT7 S1 WT in the RG assay. As the S1 ORF was replaced with the NL ORF, an additional plasmid (pCI-S1) which encodes the S1 ORF but does not contain the S1 5′ and 3′ UTRs and therefore cannot be packaged into recombinant viruses was included in the RG transfection to perform the function of the σ1 protein during virus recovery. As a negative control, the pCI-S1 plasmid was not included in the RG assay. Cell lysates were collected and viruses that packaged the NL gene during the RG assay were used to infect L929 cells. Incorporation of NL was detected by measuring NL expression in the infected L929 cells (Fig 7A). From these experiments infectious but replication-deficient virions that expressed NL luciferase were recovered when the NL ORF was flanked at each end with 200, 100, or 50 S1 nts. (Fig 7B). This was not surprising as we also recovered viruses in our W/BR assay when these sequences were WT in an otherwise wobbled S1 ORF (Fig 5B). Also, in line with our earlier results, we were unable to recover NL luciferase expressing virions using the pT7 18/NL/40 construct (Fig 7B) confirming that the S1 UTRs are insufficient for packaging the gene segment. However, unlike our previous results, in these experiments we were also unable to recover NL luciferase expressing virions when pT7 25/NL/50 or pT7 25/NL/40 plasmids were used in the RG assay (Fig 7B). This suggests that in the context of a NL ORF, 25 S1 WT nts at the 5′ end are not sufficient for packaging the

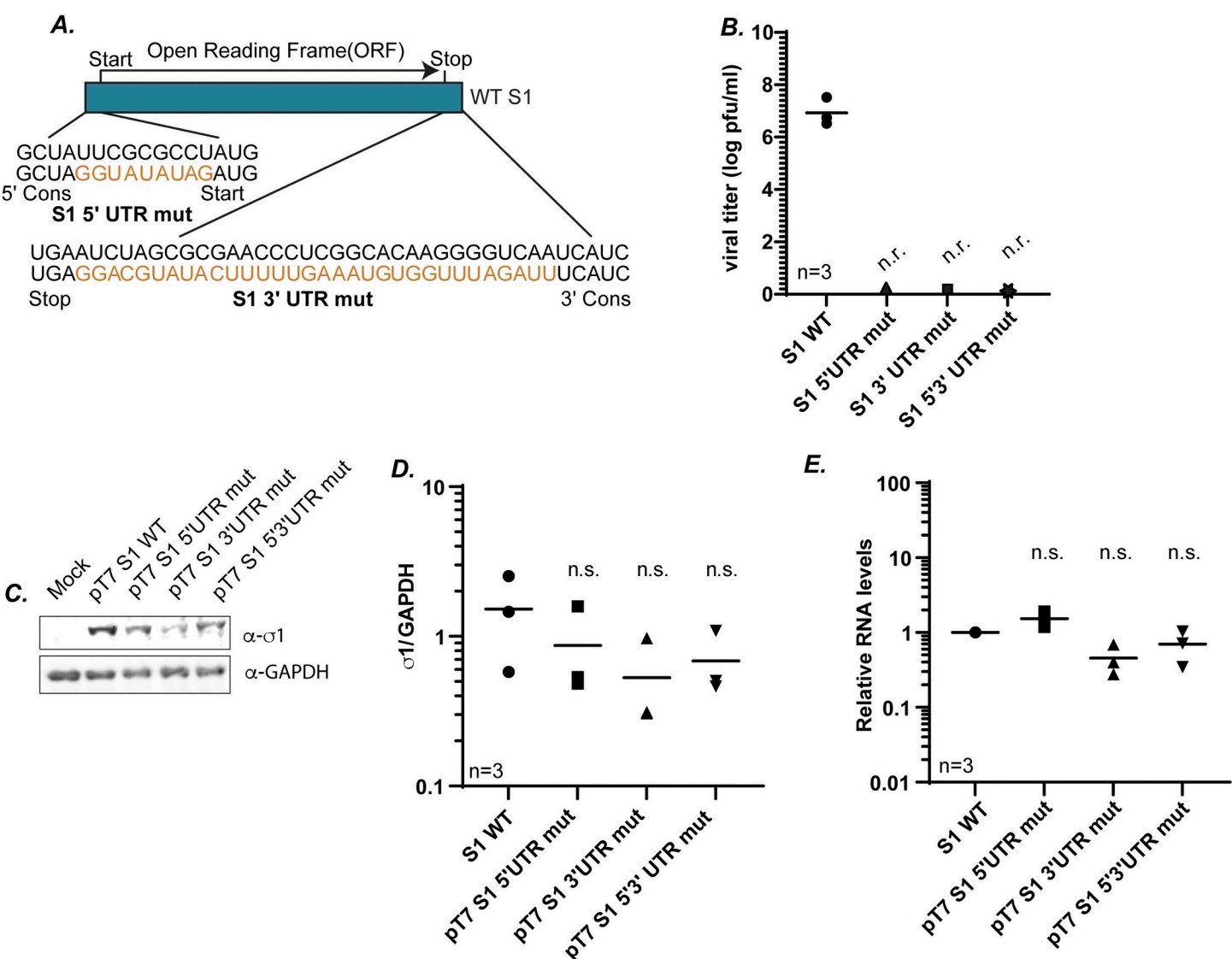

**Fig 6. 5′ and 3′ UTRs are necessary for S1 gene packaging.** (A) Illustration of UTR mutations introduced into the WT S1 gene segment construct to determine the role of UTRs in S1 gene packaging. UTR mutations are shown in orange. The nt changes were introduced into the 5′UTR (S1 5′UTR mut), the 3′UTR (S1 3′UTR mut) or both the 5′ and 3′ UTR (S1 5′3′ UTR mut). The highly conserved 5′ GCUA and 3′ UCAUC at the extreme termini were not mutated B) Viral titers (log pfu/ml) determined from RG followed by plaque assay for indicated S1 mutant gene segments. (C) Representative immunoblots of cells transfected with indicated S1 gene plasmids at 24 h post-transfection using σ1 and GAPDH antibodies. (D) Quantification of biological replicates of immunoblots in (C). σ1 protein intensities are normalized to the corresponding GAPDH loading controls. (E) Relative S1 RNA levels measured using RT-qPCR and normalized to housekeeping gene β-actin. All statistical analysis were done using one way ANOVA and Tukey multiple comparison. n.r.=not recovered, Cons=conserved, n.s.=non-significant.

segment and that WT sequences between nt 25 and nt 50 were likely contributing to rescue in the W/BR context. To address whether differences in chimeric NL virus rescue and corresponding NL luciferase expression are due to any inherent defect of NL protein expression, each of the S1/NL/S1 chimeric plasmids were transfected individually into BHK-T7 cells and measurement of NL expression showed that they expressed similar levels of NL as compared to the positive control (pNL3.1), suggesting that differences in NL expression following viral rescue cannot be attributed to differences in protein expression (Fig 7C).

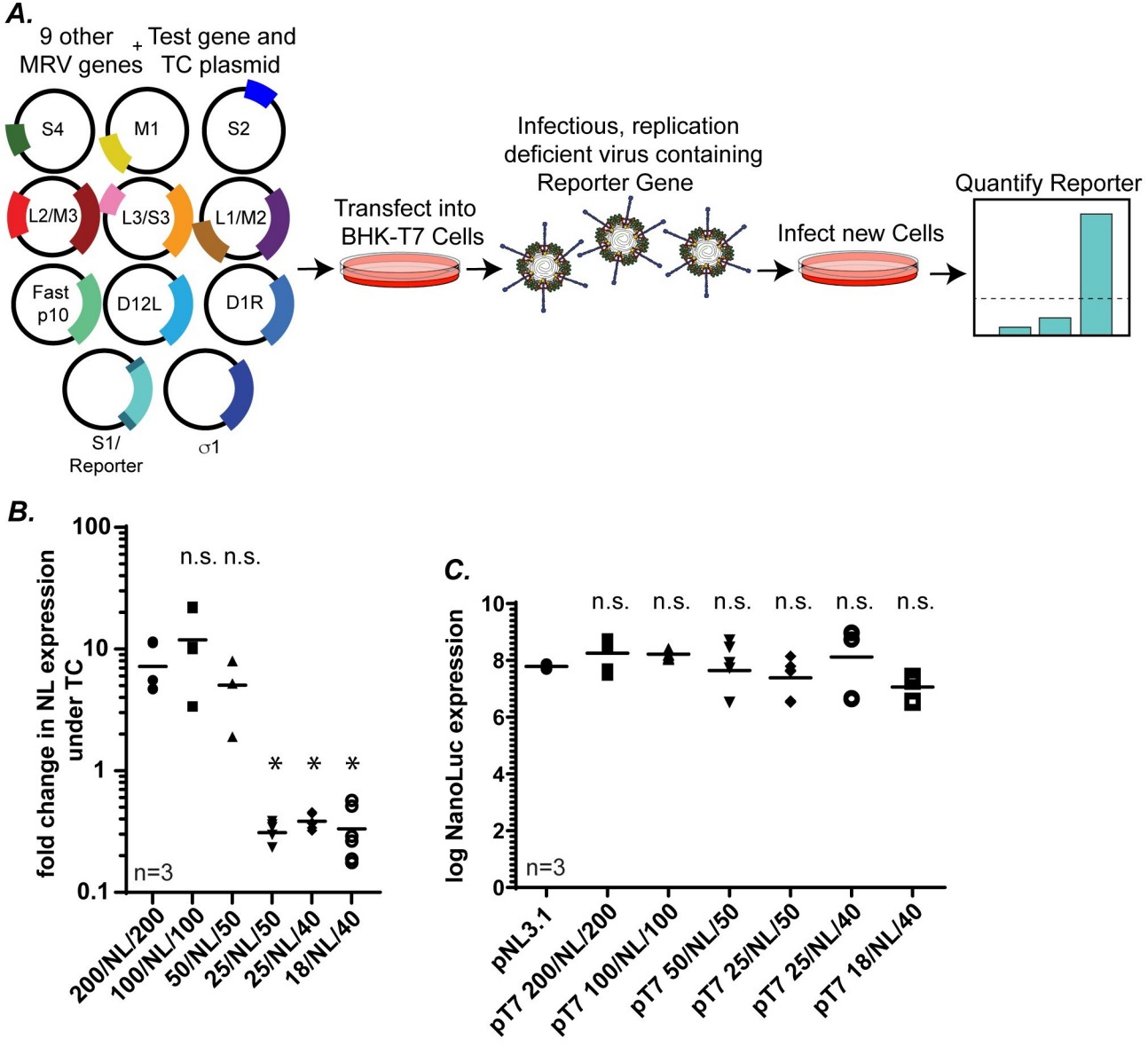

**Fig 7. 50 WT 5′ and 3′ terminal nts of the S1 gene are sufficient to package a non-MRV gene into infectious virions.** (A) Schematic of the segment incorporation assay. A chimeric S1 construct is designed (S1/ reporter) where the S1 ORF is replaced with a NL ORF that is flanked on both the ends with terminal S1 WT nts of varying lengths. This chimeric reporter plasmid is transfected into BHK-T7 cells with the other WT MRV gene segments and accessory plasmids, as well as a transcomplementing plasmid expressing the σ1 protein. 5 days post RG, rescued infectious virions are isolated and added onto new L929 cells. Incorporation of the chimeric gene segment is assayed by measurement of reporter gene expression (luciferase). (B) Segment incorporation assay using indicated S1/NL chimeric plasmids. (C) NL expression of control NL (pNL3.1) and indicated chimeric S1/NL plasmids measured 24 hours post transfection. All statistical analysis were done using one way ANOVA and Tukey comparison. *p<0.05, n.s.= non-significant, p>0.05).

## Sequences predicted to form an RNA panhandle structure between the 5′ and 3′ termini play a role in S1 packaging

In addition to defining S1 terminal sequences critical for packaging the S1 segment, we were also interested in examining the potential role of RNA structures formed by the interaction of the 5′ and 3′ ends in the packaging process. Our previous results provide evidence that the 5′ and 3′ terminal ends of S1 are both necessary for efficient virus rescue (compare rescue of 25/S1/40 with 18/S1/40 and 18/S1/50 with 18/S1/40 in Fig 5B). This suggests there may be

cooperation between the termini in the packaging process, which may present as direct interaction. It has been computationally predicted that MRV RNAs form a panhandle structure by the specific interaction of the 5′ and 3′ terminal nts [37]. However, this panhandle structure has not been verified experimentally, and its functional significance remains unclear. To begin to explore the possibility of an interaction between the 5′ and 3′ terminal ends of S1, we folded and aligned S1 gene sequences from the three major MRV serotypes using the program Multilign [38], then manually edited the outputs to align the predicted structure (Fig 8A and 8B). Two different RNA structure prediction algorithms, mfold [39] and Multilign were used, where mfold predicted single-sequence structures while Multilign was used for consensus RNA folding of multiple homologous sequences, particularly for the analysis of the panhandle

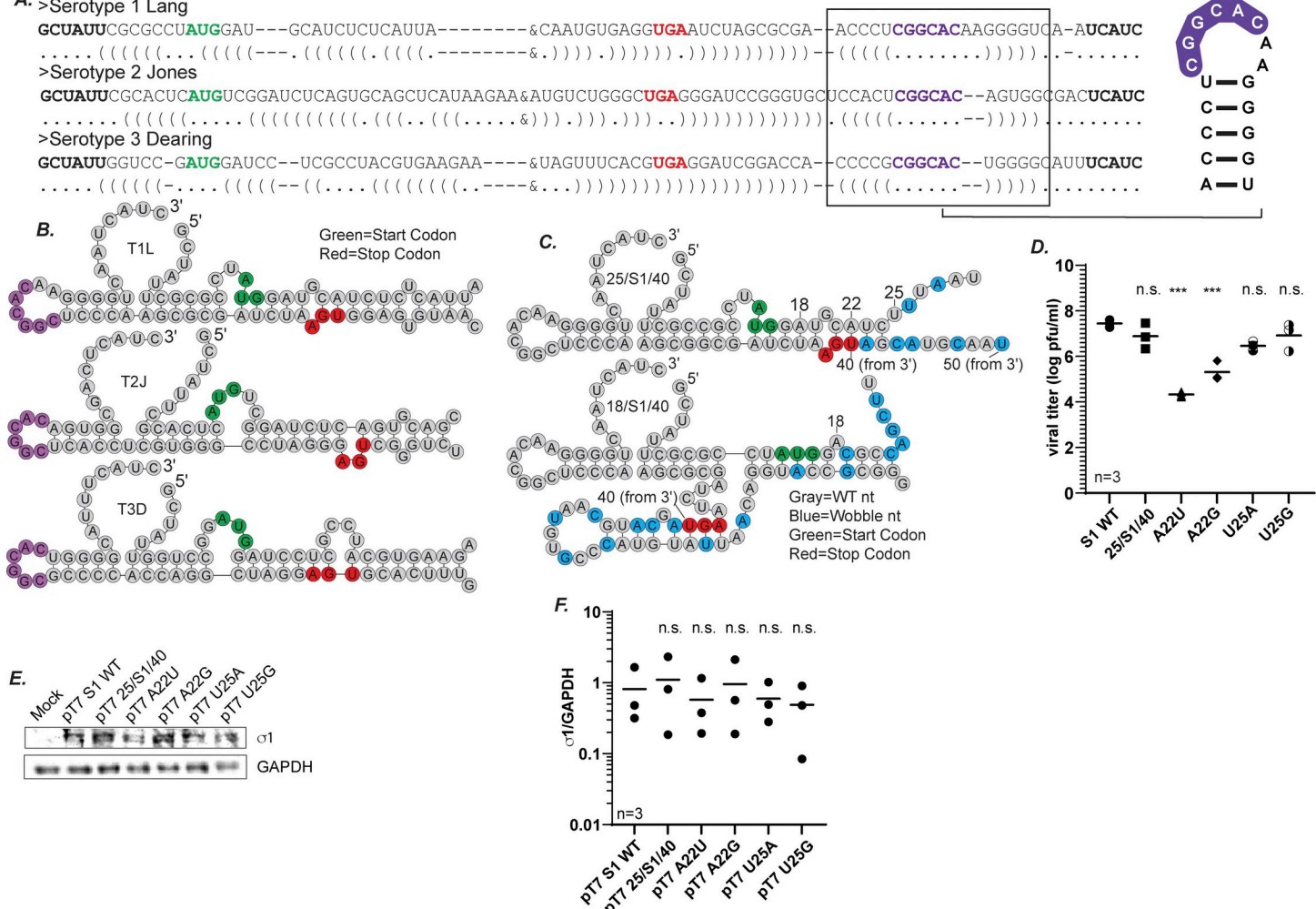

**Fig 8. Sequences predicted to form an RNA panhandle structure between the 5′ and 3′ termini play a role in S1 packaging.** (A) Dot-bracket and (B) Illustration of alignment of the terminal 5′ and 3′ end of the S1 gene from three different MRV serotypes (T1L, T2J and T3D) generated using Multialign, then manually edited to align predicted structure. A stretch of 6 conserved nts (CGGCAC) highlighted in purple form a conserved loop in the predicted panhandle structure. (C) Illustration of predicted +ssRNA secondary structures from mfold for 25/S1/40 and 18/S1/40. Start and stop codons are highlighted in green and red respectively. (D) Viral titers following RG as measured by plaque assay of the indicated S1 gene segment mutants. (E) Representative immunoblots to demonstrate protein expression from cells transfected with indicated S1 gene plasmids using σ1 and GAPDH antibodies at 24 h post-transfection. (F) Quantification of biological replicates of immunoblots in (E). σ1 protein intensities are normalized to the corresponding GAPDH loading controls. All statistical analysis were done using one way ANOVA and Tukey comparison. ***p<0.001, n.s.=non-significant, p>0.05).

structure. The program Multilign is optimized for predicting global consensus structures of RNAs that have homologous sequences and structures. It uses dynamic programming to optimize both the secondary structure and the alignment. It optimizes an alignment score and a consensus minimum free energy simultaneously and can be used to find structural homology even when sequences are quite distant. From this analysis, base pairing is possible between the 5′ and 3′ ends of the S1 gene of the three major MRV serotypes to form a structurally similar panhandle (Fig 8B). Combining the *in silico* analysis with our S1 mapping data, we hypothesized that interactions between the first 25-50 and last 40-50 nts of S1 may be required for panhandle stabilization, and that perturbation of this structure may contribute to decreased mutant rescue. When analyzed using the RNA secondary structure prediction tool mfold, we determined that the S1 mutant constructs that were rescued to WT levels from our previous results such as 25/S1/40 (Fig 5B) are predicted to form a panhandle structure identical to the panhandle structure predicted to form for the WT T1L S1 gene (Figs 8B and 8C, top). However, the 18/S1/40 construct for which we saw a significant decrease in viral recovery (Figs 2B and 5B) and plaque size (Fig 2C) had a clear change in its overall panhandle architecture with shortening of the stem structure, formation of an additional stem-loop, and changes in nt base pairing (Fig 8C, bottom).

Construct pT7 25/S1/40 differs from pT7 18/S1/40 by only seven nts. To investigate the possibility that the predicted panhandle plays a role in the packaging process, we mutated specific sequences in the 25/S1/40 construct using site-directed mutagenesis. Mutation sites were specifically chosen which were predicted to either 1) play a role in forming the panhandle stem, and therefore have an impact on the formation of the predicted panhandle (nt 22), or 2) play no role in forming the panhandle stem, and therefore have little to no impact on the predicted panhandle structure (nt 25). Two mutations each were made in nt positions 22 [pT7 25/S1/40 (A22U), pT7 25/S1/40(A22G)] and position 25 [pT7 25/S1/40(U25A), pT7 25/S1/40(U25G)] taking care to maintain the WT amino acid sequence. Each mutant plasmid was then used as a replacement for pT7 S1WT in the RG assay, followed by plaque assay to measure virus recovery. As predicted, mutations at nt position 25, which does not take part in the formation of the panhandle stem in the 25/S1/40 mfold prediction, did not change viral recovery relative to 25/S1/40 (Fig 8D). Alternatively, mutations in nt position 22 had a significant impact on viral recovery. Changing the A residue to a G led to a 1 to 1.5 log-fold reduction in viral rescue and changing the A to a U led to a 3 log-fold decrease. The difference in the efficiency of rescue between the two mutants may be due to the fact that the A to G change maintains the ability to form a G:U non canonical base pairing interaction whereas the A to U change results in a U:U in that position, which likely substantially disrupts base pairing. Nonetheless, either mutation of this predicted base pair that is likely involved in forming the 5′/3′ panhandle of S1 leads to a significant decrease in recovery suggesting that the sequences within this region, and potentially the panhandle structure itself, play an important role in the packaging process. When the plasmids were transfected into BHK-T7 cells and subjected to immunoblot analysis using antibodies against σ1, there were no significant differences in protein expression (Fig 8E and 8F), suggesting the loss in virus rescue was not a result of decreased protein expression.

## A conserved loop in the 3′ UTR plays a critical role in S1 packaging

When the full-length S1 gene sequences from the three MRV serotypes (Type 1 Lang, Type 2 Jones and Type 3 Dearing) were aligned, a stretch of 6 nts that are conserved within the 3′ UTR across all three serotypes were identified (Fig 8A). This is unusual as the S1 gene is not well conserved compared to the other MRV gene segments [37,40,41]. These six conserved nts form an unpaired loop in the predicted panhandle structure which is formed by 4/5 nt pairs

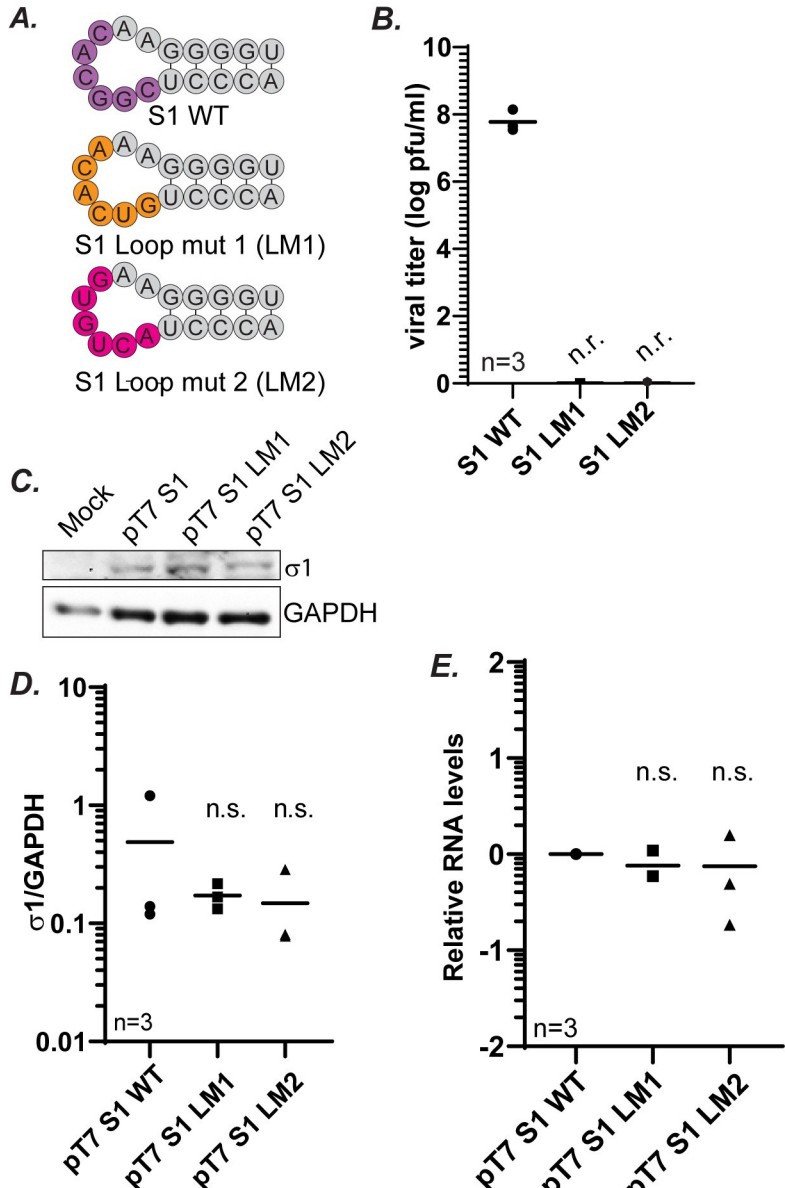

**Fig 9. A conserved loop in the 3′ UTR plays a critical role in S1 packaging.** (A) Illustration of 6 conserved nts (CGGCAC) highlighted in purple within a conserved loop in the predicted panhandle structure. Mutations that were created are indicated in orange (LM1) and pink (LM2). (B) Viral titers following RG as measured by plaque assay of the indicated S1 gene segment mutants. (C) Representative immunoblots to demonstrate protein expression from cells transfected with indicated S1 gene plasmids using σ1 and GAPDH antibodies at 24 h post-transfection. (D) Quantification of biological replicates of immunoblots in (C). σ1 protein intensities are normalized to the corresponding GAPDH loading controls. (E) Relative S1 RNA levels measured using RT-qPCR and normalized to housekeeping gene *β-actin*. All statistical analysis were done using one way ANOVA and Tukey comparison. n.r.=not recovered, n.s.= non-significant.

that contain conserved compensatory sequences across the serotypes (Figs 8B and 9A). The complete conservation across the 3 major MRV serotypes of the predicted 6 nt loop (CGGCAC) in the 3' UTR led us to hypothesize that this sequence/structure may also play an important role in S1 packaging. To investigate the role of this sequence, we constructed additional mutants CGGCAC to GUCACA (Loop mutant 1, LM1); CGGCAC to ACUGUG (Loop

mutant 2, LM2) to change the conserved sequence without modifying the predicted panhandle/loop structure (Fig 9A). These loop mutant plasmids were used in place of pT7 S1 WT in RG assays and cell lysates were subjected to plaque assay to determine virus titer. We were unable to rescue any replicating viruses when using these mutant plasmids, strongly suggesting the conserved sequence within the loop has an important function in S1 packaging (Fig 9B). To rule out whether the introduced mutations resulted in changes in protein expression that led to a loss of viral rescue, the pT7 S1 LM1 and pT7 S1 LM2 plasmids were transfected into BHK-T7 cells, and at 24 h post transfection, immunoblotting experiments were performed. Our data showed no significant differences in σ1 protein expression (Fig 9C and 9D). Additionally, when relative RNA levels were measured using RT-qPCR, there were no significant differences in gene expression of S1 between the mutant constructs and WT S1 (Fig 9E). Altogether, these results suggest that changing six highly conserved nts that are present in a predicted unpaired loop in an otherwise WT S1 gene leads to a complete loss of recovery of replicating viruses. These findings strongly suggest that this sequence within the unpaired loop plays a critical role in packaging the S1 gene segment.

## Differences in viral rescue for S1 UTR mutants cannot be attributed to differences in σ1 protein expression

Our previous results show that the mutations in the UTR of the S1 gene (S1 5′UTR mut, S1 3′UTR mut, S1 5′3′UTR mut, S1 LM1, S1 LM2) leads to loss of recovery of replicating virus, suggesting the UTR plays a role in the packaging process (Figs 6B and 9B). While we provide evidence that mutation of these sequences do not result in loss of RNA or protein expression, in order to alleviate all concerns that differences in viral recovery could be attributed to differences in expression of σ1, we performed a σ1 transcomplementation assay. In this assay we performed RG replacing pT7 S1 WT with the S1 UTR mutant constructs (pT7 S1 5′UTR mut, pT7 S1 3′UTR mut, pT7 S1 5′3′UTR mut, pT7 S1 LM1, and pT7 S1 LM2). In addition a pCI-S1 plasmid which encodes only the σ1 ORF without 5′ and 3′ UTRs and will perform the function of the σ1 protein during virus recovery, was included in the RG transfection. Three days post RG under σ1 transcomplementation, viral titer was determined by plaque assay on L929 cells. Samples were additionally subjected to immunoblot using σ1-specific antibodies to measure σ1 protein levels across RG samples. Consistent with our prior experiments without σ1 transcomplementation (Figs 6B and 9B), results from this assay show no virus rescue in the UTR mutant constructs (Fig 10A), even in the presence of similar expression of σ1 across all the constructs tested (Fig 10B). These results provide additional evidence that differences in viral recovery of the UTR mutants relative to WT S1 cannot be attributed to differences in expression of σ1 protein.

## Discussion

Despite decades of research, very little is understood about the mechanisms driving the packaging process of MRV. In this work, we have designed a novel assay to aid in defining viral packaging in the context of replicating viruses through introduction of wobble codons throughout the ORF. Using this approach, sequences sufficient for packaging the T1L S gene segments (S1-S4) and the minimal nt sequences sufficient for packaging the S1 gene have been identified for the first time. An RNA secondary structure was predicted computationally to exist by base pairing of the terminal ends of the MRV genome segments [37]. Our data lend experimental support that this predicted secondary structure may play an important role in MRV genome packaging. Furthermore, we have identified a stretch of six nts conserved across MRV serotypes that are part of an unpaired loop within the panhandle structure as critical to

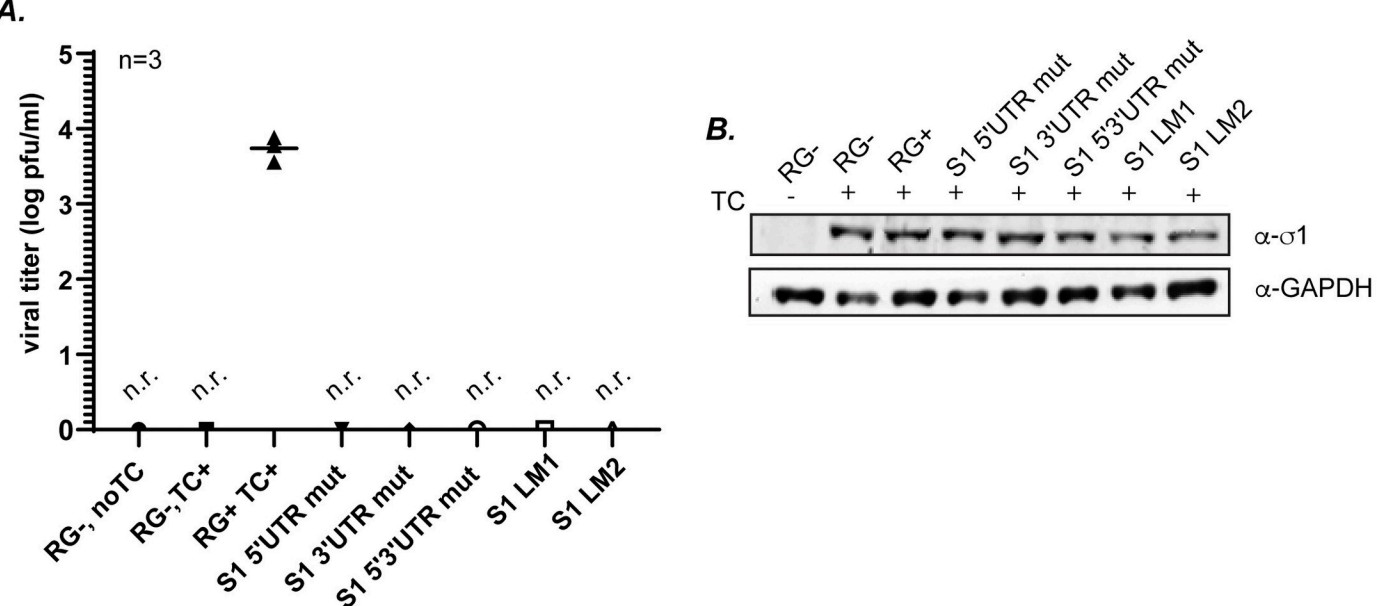

**Fig 10. Differences in viral rescue for UTR mutants cannot be attributed to differences in translation of σ1 protein** (A) Viral titer of indicated S1 UTR mutants following RG assay transcomplemented with a plasmid expressing WT σ1 as measured by plaque assay of RG lysates on L929 cells. (B) Representative immunoblot of σ1 protein measured 3 days post RG in indicated transcomplementation assays using σ1 antibodies. GAPDH immunoblot was used as a loading control. TC=transcomplementation, RG-, Reverse genetics negative control (no pT7 S1). RG+: Reverse genetics positive control (pT7 S1 WT), n.r.=not recovered.

the packaging process. The information we have gained from our research reveals new details about the MRV packaging process and leads to many additional exciting questions to be addressed in future research.

We have identified that the packaging sequences for the T1L S gene segments are contained within the terminal 50-200 nts of the gene inclusive of the UTRs. In future work, we would like to determine the specific functional significance of these packaging signals and clearly identify which step in the packaging process (assortment, assembly, replication, or more than one of the processes) these sequences are required. All MRV gene segments have a conserved GCUA at the 5′ UTR and a UCAUC at the 3′UTR [27] that likely functions as the binding site for the MRV RdRP protein λ3, and is essential for replication and transcription of mRNA [36]. In the mutant constructs included in this study, these conserved residues were not mutated, therefore it seems less likely that λ3 binding, second-strand synthesis or mRNA synthesis have been impacted in our studies. Recent studies also suggest the polymerase of dsRNA viruses are required to bring the assortment complex to the assembling virion [24,42]. Again, because we have not altered the putative λ3 binding site in our mutants, it also seems unlikely our mutants would alter the core assembly process. For these reasons we believe the most likely explanation for the decreased/loss of virus rescue of many of these mutants is that they disrupt RNA:RNA interactions within and between gene segments and interfere with assortment of the MRV genome. Confirmation of this hypothesis will depend on the development of an MRV assortment-specific assay to demonstrate that these mutants are defective in assortment and not some other stage of packaging.

Because the mutations we introduced into the S genes had the potential to impact steps in the virus life cycle prior to the packaging process we carefully examined these steps throughout the manuscript. First, to address developing concerns that the introduction of wobble mutations into coding regions may lead to decreased levels of gene expression [43] and the possibility that our UTR mutations may impact protein expression, we performed immunoblot

analysis on protein expression of each of our mutants to show that levels of σ protein expression are generally not significantly different between WT and wobble mutant constructs (Figs 3A–3D, 5C–5D, 6C–6D, 8E–8F, 9C–9D). We also examined RNA expression from most of our mutants to rule out any impacts of mutations on RNA (Figs 3G, 5E, 6E, 8G, 9E). Additionally, for some of our mutants we performed transcomplementation assays, where a plasmid expressing WT protein is provided in trans in RG assays, which provide further evidence that the inability to recover virus from RG assays is not a result of less protein expression (Figs 3E–3F and 10). Further, because prior studies have found that at least 2 MRV proteins may play a role in regulation of viral protein translation [44,45], we measured protein expression in cells transfected with S gene segment plasmids together with the remaining 9 MRV gene segments on a subset of genes and found that there were no significant changes in the expression of these proteins (Fig 3H) compared to that observed when the plasmids were expressed individually (Fig 3A and 3C), further supporting that our inability to rescue viruses in RG assays are not due to differences in protein expression. Moreover, we showed that the 18/S1/40 virus is able to enter cells, and transcribe and translate proteins (Fig 3J), but is impaired in replication as demonstrated by the small plaque phenotype (Fig 2C), replication assay (Fig 3I), and immunofluorescence assay (Fig 3J) again demonstrating that the defect introduced by this mutation is occurring at a later time in the viral life cycle. Taking this data altogether, we can safely conclude that introduction of wobble mutations across the ORF as well as our targeted mutations in the S gene UTRs likely impact a step during the packaging process of the viruses and are not the result of insufficient available proteins or RNA for replicating virus production. However, the possibility remains that mutants that we were unable to recover in our plaque assays may produce non-infectious virus with defects in processes needed to infect the next round of cells.

Another question of interest is whether sequences required for packaging are located exclusively at the terminal ends of the MRV segments. Our data shows individual S gene segments, and a combination of all four S gene segments with 200 WT terminal nts flanking wobbled sequence can be packaged into replicating virus. Moreover, when just the UTRs are WT and the remaining ORF is wobbled, the S gene segments cannot be efficiently packaged. This data provides strong evidence that MRV packaging signals are localized to specific sequences/structures at the gene termini, and that there is limited flexibility within the MRV genome to compensate for mutations made within the packaging sequences. This would be in contrast to another segmented virus, Influenza A, which shows high flexibility of binding between gene segments during packaging [46]. In support of this idea of limited flexibility in MRV packaging, we also show that mutations within the UTRs result in total loss of viral recovery, even though all the nts in the ORF remain WT. This suggests that disruption of these sequences, or the RNA structures they form, cannot be compensated by other sequences elsewhere within the genome. Moreover, addition of just 50 S1 terminal nts onto a non-MRV gene is sufficient to package that gene into replicating virus. Taken altogether, this data may suggest that MRV (and other dsRNA viruses) have a static RNA interaction network that is heavily sequence specific and/or structure driven. If this is the case, it seems likely that some level of flexibility must be present to account for the known property of reassortment between virus gene segments that occurs in cells infected with multiple MRV serotypes.

It is possible that MRV genome assortment is driven by a central gene segment that interacts with all other gene segments. For BTV and RV, the smallest ssRNA segment has been speculated to initiate the complex network of RNA:RNA interactions, with the UTRs of the smallest segments necessary for the process [12,14,47] So far with our results, we have not provided evidence that supports or refutes the possibility that S gene segments play a central role in interacting with the other gene segments. Interestingly, mutation of the conserved unpaired loop in the 3′ end of S1 resulted in complete loss of virus rescue. This loss of rescue may

suggest that these sequences form a specific interaction with complementary sequences in other gene segments as part of the assortment process. In support of this hypothesis, the 3′ terminal end of avian reovirus S1 ssRNA segment hybridizes with avian reovirus S4 ssRNA segment *in vitro* [48]. In a scan of MRV T1L segments, we found 4 genes (S4, M2, L1, and L3) that contained exact complementary sequences which could be possible RNA:RNA binding partners to this loop within S1 during assortment. It is also possible that all 6 nts are not involved in the RNA:RNA interaction network formed during assortment, that base pair interactions that occur between these 6 nts and another gene segment are not linear in nature, or that this sequence plays a necessary, but not sufficient role in S1 gene segment assortment or the overall packaging process. Nonetheless, if these sequences are involved in joining S1 to the assortment complex, it seems unlikely that S1 plays a central role in assortment where it recruits more than one gene segment to the complex through binding the unpaired loop. While we provide clear evidence that 200 nts at the termini of the other S gene segments are sufficient for their incorporation into the assortment complex and/or other steps in the packaging process, additional fine mapping experiments will have to be performed to clarify specific sequences/structures within the termini that are necessary for this process. Moreover, while prior studies have shown that the M and L gene segment termini are also likely to be important in packaging, future experiments in our and other laboratories will be necessary to determine the mechanism of MRV gene segment assortment.

RNA folding algorithms predict that the terminal sequences in each of the 10 MRV genes form a panhandle structure through interaction of the terminal ends [37]. The mfold computational prediction for the S1 gene also suggests that the determined packaging signal sequences (5′ 25 and 3′ 50) identified in this study interact as part of the panhandle structure. Our data lend experimental proof that mutants that are predicted to modify the panhandle lead to reduced viral recovery. In addition to the panhandle, the rest of the MRV gene is also predicted to be highly structured. It is possible that the introduction of wobble mutations in the ORF alters the +ssRNA structure such that internal regions of the ORF no longer form meaningful interactions necessary for some step in the packaging process, such as interaction with the other gene segments. However, our data alleviates that concern because many of the constructs used throughout this study are wobbled across much of the gene segment, yet are rescued to WT levels, suggesting that inter- or intrasegment folding parameters needed for the packaging process are maintained. In fact, the 25/S1/40 virus, which maintains only 25 WT nt sequences at the 5′ end and 40 nts at the 3′ end, is recovered to WT virus levels even though most of the gene is wobbled. It is only when changes are made in the predicted panhandle structure that viral recovery is negatively impacted, further showing the importance of the panhandle structure or sequences contained within the panhandle region. Although our data suggest that at least for the S segments RNA sequences or structures internal to the terminal 200 nts are not necessary for packaging of these segments into the MRV virion, we have not yet investigated the larger M or L segments and it remains quite possible that RNA structure beyond the panhandles, both within and between gene segments plays an important role in MRV packaging.

## Materials and methods

### Cells

BHK-T7 cells expressing the T7 polymerase [49] and Vero cells were maintained in Dulbecco′s modified Eagle′s medium (DMEM) (Invitrogen Life Technologies) supplemented with 10% fetal bovine serum (Atlanta Biologicals), Penicillin (100 IU/ml)–Streptomycin (100 μg/ml) solution (Mediatech), and 1% MEM nonessential amino acids solution (HyClone). To maintain the expression of T7 polymerase in BHK-T7 cells, 1 mg/ml of G418 (Alexis Biochemical)

was added after every four passages. Spinner adapted L929 cells were maintained in Joklik modified minimum essential medium (Sigma-Aldrich) supplemented with 2% fetal bovine serum, 2% bovine calf serum (HyClone), 2 mM L-glutamine (Mediatech), and penicillin (100 IU/ml)–streptomycin (100 μg/ml) solution.

## Plasmid construction

The ten [50] (ID: 33295, 33294, 33293, 33292, 33291, 33290, 33289, 33288, 33287, 33286) and four [51] (ID: 33306, 33305, 33304, 33303) reverse genetics (RG) system plasmids for T1L, along with plasmids encoding the vaccinia virus capping enzymes pCAG D1R (ID:89160) and pCAG D12L(ID:89161) and a Nelson Bay Virus syncytia inducing plasmid pCAG Fast p10 (ID:89152) [52] were purchased from Addgene. pCI-S2 (T1L) was previously described [53]. pCI-S1 is a plasmid expressing Type 3 Dearing σ1 and was a kind gift from Dr. Max Nibert. To construct pT7 S1 WT, pT7 S2 WT, pT7 S3 WT, pT7 S4 WT, first a pCI-Rz plasmid was created by PCR amplification of the Hepatitis δ ribozyme sequence and ligation into the multiple cloning site of the pCI-Neo plasmid (Promega) using 5′ SacI/3′ SmaI restriction endonuclease sites. WT T1L S gene segment plasmids were generated by RT-PCR amplification of viral S1, S2, S3, and S4 gene segments respectively from purified T1L virus with primers that contain a 5′ SacI cleavage site upstream of a T7 promoter sequence. The 5′ SacI-T7-S gene segment 3′ PCR product was then ligated into the pCI-Rz plasmid using the following cleavage sites: S1: 5′ SacI/3′ XhoI (partial restriction digestion as S1 ORF has an internal SacI site), S2: 5′ SacI/3′ AvaI, S3: 5′ SacI/3′ NheI, S4: 5′ SacI/3′ XbaI.

The wobble mutated S gene and S1/NL gene segments were synthesized as double-stranded (ds) DNA sequences (gBlocks) by Integrated DNA Technologies with restriction endonuclease cleavage sites at the 5′ and 3′ end of the dsDNA. The gBlocks were digested and cloned into the parent gene segment plasmids (pT7 S1WT, pT7 S2WT, pT7 S3WT, pT7 S4WT) following the NEB Cloner workflow. Specific mutant constructs were subcloned from the mutants using appropriate restriction digestion sites or using the site-directed mutagenesis kit (Quick Change II Site directed mutagenesis kit, Agilent Technologies) following manufacturers protocol. The 5′ and 3′ restriction endonuclease sites, vector backbones, and primer sequences used in site directed mutagenesis for making the mutant S gene segment constructs are listed in Table 1.

Table 1 shows the mutant S gene segment plasmid constructs, with the restriction endonuclease cleavage sites used at the 5′ end (5′ RE) and the 3′ end (3′ RE) along with the vector backbone used for cloning in the insert, and the primers (Forward primer: FP and Reverse primer: RP) used during site directed mutagenesis (SDM).

## Infections and transfections

Our laboratory stock of T1L is derived from the lab of B.N Fields. Viruses were purified using a method described previously [54] using Vertrel-XF (Miller Stephenson Chemical company, Catalog: NC9715008) instead of freon [55]. Multiplicity of infection calculations for T1L coinfections in the Segment Incorporation Assay were based on cell infectivity unit (CIU) measurement as described previously [56–58]. Transfections were carried out in Opti-MEM (Thermo Fisher Scientific) using TransIT-LT1 transfection reagent (Mirus, Cat: MIR 2306) following the manufacturer's protocol with a TransIT-LT1(μl): plasmid (μg) ratio of 3:1.

## Antibodies

σ2 rabbit polyclonal antiserum against heat inactivated core [59], rabbit polyclonal σNS antiserum [60], and rabbit anti T1L virion serum for σ3 [61,62] were used for immunoblot assays. Rabbit (σ1) VU304 antisera raised against the T1L σ1 head domain was a generous gift from

**Table 1. Mutant S gene segment plasmid design.**

| Plasmid | 5′ RE | 3′ RE | Vector Backbone | Primer sequence (5′-3′) |
|---|---|---|---|---|
| pT7 200/S1/200 | BbsI | AgeI | pT7 S1WT | |
| pT7 100/S1/100 | Subcloned from pT7 100/S1/40 and pT7 18/S1/100 | | | |
| pT7 50/S1/50 | Subcloned from pT7 50/S1/40 and pT7 18/S1/50 | | | |
| pT7 100/S1/40 | SpeI | XhoI | pT7 18/S1/40 | |
| pT7 50/S1/40 | SpeI | XhoI | pT7 18/S1/40 | |
| pT7 25/S1/40 | SpeI | XhoI | pT7 18/S1/40 | |
| pT7 18/S1/100 | XhoI | Not I | pT7 18/S1/40 | |
| pT7 18/S1/50 | XhoI | Not I | pT7 18/S1/40 | |
| pT7 18/S1/40 | SpeI | NotI | pT7 S1 WT | |
| pT7 200/S2/200 | SacI | NotI | pT7 S2 WT | |
| pT7 23/S2/59 | SacI | NotI | pT7 S2 WT | |
| pT7 200/S3/200 | SacI | NotI | pT7 S3 WT | |
| pT7 32/S3/73 | SacI | NotI | pT7 S3 WT | |
| pT7 200/S4/200 | SpeI | NotI | pT7 S4 WT | |
| pT7 37/S4/69 | NdeI | NotI | pT7 S4 WT | |
| pT7 25/S1/40 (A22U) | SDM | | pT7 25/S1/40 | **FP:** attcgcgcctatggatgcttctttaatcaccgaaatc<br>**RP:** gatttcggtgattaaagaagcatccataggcgcgaat |
| pT7 25/S1/40 (A22G) | SDM | | pT7 25/S1/40 | **FP:** attcgcgcctatggatgcgtctttaatcaccgaaatc<br>**RP:** gatttcggtgattaaagacgcatccataggcgcgaat |
| pT7 25/S1/40 (U25A) | SDM | | pT7 25/S1/40 | **FP:** gcgcctatggatgcatcattaatcaccgaaatcag<br>**RP:** ctgatttcggtgattaatgatgcatccataggcgc |
| pT7 25/S1/40 (U25G) | SDM | | pT7 25/S1/40 | **FP:** gcgcctatggatgcatcgttaatcaccgaaatcag<br>**RP:** ctgatttcggtgattaacgatgcatccataggcgc |
| pT7 S1 5′UTR mut | SDM | | pT7 S1 WT | **FP:** acgactcactatagctaggtatatagatggatgcatctctca<br>**RP:** tgagagatgcatccatctatatacctagctatagtgagtcgt |
| pT7 S1 3′UTR mut | AgeI | NotI | pT7-S1 WT | |
| pT7 S1 5′3′ UTR mut | Subcloned from pT7S1 5′UTR mut and pT7S1 5′3′UTR mut | | | |
| pT7 S1 LM1 | SDM | | pT7 S1 WT | **FP:** ccgacccgatgattgaccccttttgtgacagggttcgcgctagattcacct<br>**RP:** aggtgaatctagcgcgaaccctgtcacaaaggggtcaatcatcgggtcgg |
| pT7 S1 LM2 | SDM | | pT7 S1 WT | **FP:** cgacccgatgattgaccccttcacagtagggttcgcgctagattcacc<br>**RP:** ggtgaatctagcgcgaaccctactgtgaaggggtcaatcatcgggtcg |
| pT7 200/NL/200 | SacI | NotI | pT7 S1 WT | |
| pT7 100/NL/100 | SacI | NotI | pT7 S1 WT | |
| pT7 50/NL/50 | SacI | NotI | pT7 S1 WT | |
| pT7 25/NL/50 | SDM | | pT7 18/NL/40 | **FP:** aacgcattctggcgtaacaatgtgaggtgaatctagcgcgaaccct<br>**RP:** agggttcgcgctagattcacctcacattgttacgccagaatgcgtt |
| pT7 25/NL/40 | SDM | | pT7 18/NL/40 | **FP:** tatagctattcgcgcctatggatgcatctatggaagctcgacttcc<br>**RP:** ggaagtcgagcttccatagatgcatccataggcgcgaatagctata |
| pT7 18/NL/40 | SacI | NotI | pT7 S1 WT | |

the Dermody lab, University of Pittsburgh. Rabbit polyclonal antiserum µNS [63] and donkey anti-rabbit Alexa Fluor 488 secondary antibody (Thermo Fisher Scientific, Cat: A21206) were used for the immunofluorescence assay. Mouse monoclonal GAPDH (Thermo Scientific, Cat: MA5-15738) and rabbit α-TUBULIN (Cell Signaling Technologies, Cat: 50-190-297) were used as loading controls in immunoblot assays.

## Wobble/Block Replacement (W/BR) assay

BHK-T7 cells were plated overnight at $2x10^5$ in a 12 well dish. 0.4 µg of plasmids from the MRV four and ten plasmid RG system [50,51] were transfected to express the nine WT MRV gene segments. 0.4 µg of the mutant gene segment was transfected along with the above plasmids to provide the tenth segment.

A modification was made from the original four plasmid RG system protocol [50] by the addition of plasmids expressing vaccinia virus capping enzymes pCAG D1R (0.2 µg), pCAGD12L (0.2 µg) and Nelson Bay Virus fusion associated transmembrane protein pCAG-Fast p10 (0.005 µg) that have been previously reported to enhance viral recovery in RG by almost 1,150 fold [52]. All RG transfections were allowed to progress for 5 days unless otherwise mentioned, after which cells were subjected to three freeze-thaw cycles to recover virus.

## Plaque assay

Viral titers of replicating viruses were determined using standard plaque assay in L929 cells [54]. L929 cells were plated in 6 well plates at $1.5x10^6$ per well. After overnight incubation, viral samples were serially diluted in Phosphate Buffered Saline (PBS) $MgCl_2$ [0.2 mM $MgCl_2$ supplemented to 1X PBS (137 mM NaCl, 3 mM KCl, 8 mM $Na_2HPO_4$, 1.5 mM $KH_2PO_4$, pH 7.4)]. Viral dilutions were added onto the L929 monolayers at room temperature for viral attachment for 1 hour. Post-viral attachment, an overlay media consisting of 2X Serum Free Medium 199 (Gibco) supplemented with 1X Penicillin Streptomycin and 1X L-glutamate was added along with 2 mg/ml Trypsin (final concentration: 80 µg/ml), 2% Agar (1% final concentration) and 5% Sodium Bicarbonate (0.15% final concentration). Viral titers were determined by counting plaques three or four (for experiments including 18/S1/40) days post overlay addition.

## Crystal Violet staining of plaques

After determination of viral titer, 8% paraformaldehyde was added to each well to fix the L929 cells and left to incubate overnight. The next day, overlay media and paraformaldehyde were gently removed and cells were washed once with 1X PBS. 1% Crystal Violet in 20% ethanol was added and incubated at room temperature for 15-30 min to stain the fixed L929 cells, after which the staining solution was removed and left at room temperature for 1 min. Plaque images were taken using an iPhone 13 ProMax camera set-up that maintained the plates at the same distance from the camera and plaque sizes were measured using ImageJ.

## Viral RNA extraction and sequencing

To extract viral RNA of recovered viruses from RG, individual plaques, picked using disposable glass Pasteur pipettes (Fisherbrand), were added to 1 ml of media and subjected to 3X freeze/thaw cycles (Passage 0). For passage 1, each 1ml passage 0 sample was added to a T25 flask of L929 cells plated at $2.5x10^6$ L929 cells and infection was left to proceed until 70-80% cytopathogenicity was observed. Thereafter, the passage 1 T25 flask was freeze thawed 3X. For passage 2, 500µl-1ml of passage 1 was added to a T75 flask plated with $7.5x10^6$ L929 cells or $2.5x10^6$ Vero cells (for 18/S1/40) and infection was left to proceed until 70-80% cytopathogenicity was observed. The passage 2 flask was freeze thawed 3X and viral RNA was extracted using TRIzol LS (Life Technologies) following manufacturer's instructions. Approximately 1µg of total RNA was reverse transcribed using SuperScript IV (Invitrogen Life Technologies) as per manufacturer's protocol. cDNA was PCR amplified using gene specific primers to produce full-length gene segments, and the entire genes were sequenced using gene specific primers.

## RT-qPCR assay

$5x10^5$ BHK-T7 cells were plated in 6 well dishes and after overnight incubation, transfected with 3μg of total plasmid using TransIT-LT1 transfection reagent (Mirus, Cat: MIR 2306) following the manufacturer's protocol with a TransIT-LT1 (μl): plasmid (μg) ratio of 3:1. 24 hours post-transfection, cells were collected in PBS and total RNA was extracted using Pure-Link RNA miniprep kit (Invitrogen Life Technologies) following manufacturer's protocols. Total RNA was then digested using DNase I for 2h at 37˚C (New England Biolabs) to remove residual plasmid DNA and then purified using Monarch RNA cleanup kit (New England Biolabs). RNA was reverse transcribed using SuperScript IV (Invitrogen Life Technologies) or qScript Ultra Flex Kits (Quantabio), using gene specific primers (Integrated DNA Technologies). The real time PCR was performed using PowerTrack SYBR Green Master Mix (Applied Biosystems) and SYBR Green detection was done using QuantStudio3 Real-Time PCR system. The Real-Time PCR was done using standard cycling conditions [Enzyme activation: 95˚C for 2 min, (Denaturation: 95˚C for 15s, Anneal/Extend: 60˚C for 60s) for 40 cycles]. Melt curve analysis was done at 95˚C for 15s, followed by 60˚C for 60s. The dissociation analysis was done at 95˚C for 15s. Data analysis was performed using Design and Analysis software 2.6.0 (Applied Biosystems). The data is reported using the $2^{-\Delta\Delta CT}$ method of relative quantification [64]. Briefly the $C_T$ value of the S segment PCR was subtracted from the $C_T$ value of the housekeeping gene PCR to generate a normalized $C_T$ value for the S segment. This normalized $C_T$ value was then subtracted from the normalized $C_T$ value of the reference gene to generate a $\Delta\Delta C_T$ value. This $\Delta\Delta C_T$ value was then used in the formula $2^{-\Delta\Delta CT}$ and compared to the reference gene for reporting the relative RNA level.

## Segment incorporation assay

Chimeric S1 NL plasmids and the pCI-Neo S1 plasmid were used to replace pT7-S1 in the RG assay. Two days post RG assay, media was removed, samples were washed 3 times with 1X PBS and then refed with fresh DMEM to remove untransfected plasmids from the samples. The RG reaction was allowed to proceed an additional 3 days, after which samples were subjected to three freeze-thaw cycles to release virions from cells. The cells debris was removed after centrifugation at 15000g for 5 mins and cell supernatant containing the released virions were subjected to 5 μl DNase I (Zymo research) for 1 hour to remove remaining untransfected chimeric NL plasmid. Equal volumes of DNase I treated samples were then co-infected with WT T1L into L929 cells plated in 12 well dishes at $2x10^5$ cells per well. DNaseI treated samples were coinfected with WT T1L to support additional rounds of NanoLuc/S1 gene packaging and corresponding infection for better detection of the NanoLuc signal from the infected cells. Twenty-four hours post-infection, cells were gently washed and collected in sterile 1XPBS. The cells were treated with NanoGlo Luciferase reagent (Promega) according to manufacturer's protocol and Luciferase activity was measured using a Glomax Multi Detection Plate reader on a 96 well plate.

## Immunoblot

$5x10^5$ BHK-T7 cells were plated overnight on 60mm dishes and transfected with 10μg of individual plasmids in BHK-T7 cells and protein expression was measured 24 hours post transfection. Twenty-four hours post-transfection, cells were collected in ice cold 1XPBS and lysed with RAF buffer (20 mM Tris [pH 8.0], 137 mM NaCl, 10% glycerol, 1% NP-40) [65] for 1 hour on ice. For measurement of protein expression of S1 and S3 genes in the presence of 9 other RG MRV gene segments and accessory plasmids, BHK-T7 cells were cultured in either 100 mm or 60 mm tissue culture dishes until 40-60% confluency was obtained. The cells

grown in 100 mm dishes were transfected with 3 μg of each RG plasmid, 1.5 μg of pCAG D1R, 1.5 μg of pCAGD12L, and 0.0375 μg of pCAG D1R. The cells cultivated in 60 mm dishes were transfected with 1.2 μg of each RG plasmid, 0.6 μg of pCAG D1R, 0.6 μg of pCAGD12L, and 0.015 μg of pCAG D1R. After 24 h of transfection, the cells were washed with PBS, and centrifuged to collect the cell pellet. Cell pellets or whole cell lysates in RAF were resuspended in 2X Laemeli loading dye (125 mM Tris-HCl [pH 6.8], 200 mM dithiothreitol [DTT], 4% sodium dodecyl sulfate [SDS], 0.2% bromophenol blue, 20% glycerol), and subjected to sodium dodecyl sulfate-polyacrylamide gel electrophoresis (SDS-PAGE) for protein separation. Membranes were blocked using 5% Milk in TBS-T (20 mM Tris, 137 mM NaCl [pH 7.6]) with 0.1% Tween 20 and treated with primary and secondary antibodies for 18 h and 2 h respectively with 3x15 minute TBS-T washes after each antibody incubation. Membranes were treated with Nova-Lume Atto Chemiluminescent Substrate AP (Novus Biologicals) using manufacturers protocol and imaged on a ChemiDoc XRS Imaging System with Quantity One imaging software (Bio-Rad Laboratories).

### Immunofluorescence assay

$1x10^5$ BHK-T7 cells were grown on 18mm glass coverslips in 12 well plates (Fisherbrand) overnight and infected with S1 WT, 200/S1/200 and 18/S1/40 viruses at an MOI of 0.001. 24h post-transfection, cells were fixed with 4% paraformaldehyde for 20 min, permeabilized with 0.2% Triton X-100 in PBS for 5 min and blocked with 1% bovine serum albumin in 1X PBS (PBSA) for 30min with 1X PBS washes between each step. Primary antibody (μNS rabbit polyclonal antiserum) [63] and secondary antibody (donkey anti-rabbit Alexa Fluor 488) were diluted in PBSA and cells were incubated with primary and secondary antibodies for 45 mins at room temperature with three 1X PBS washes between each step. Coverslips were mounted using ProLong Gold Antifade reagent with DAPI (4,6-diamidino-2-phenylindole dihydrochloride; Invitrogen Life Technologies). Images were captured using Zeiss Axiovert 200 inverted 90 fluorescence microscope at 100X objective lens under oil immersion and processed using Adobe Photoshop and Illustrator software.

### Protein transcomplementation assay

0.4 μg S segment mutant constructs (pT7 23/S2/59, pT7 S1 5′UTR mutant, pT7 S1 3′UTR mutant, pT7 S1 5′3′ UTR mutant, pT7 S1 LM1, or pT7 S1 LM2) were subjected to the RG procedure as described above. σ protein was provided by transfecting 0.4 μg of pCI-S1 (for S1 5′UTR mutant, S1 3′UTR mutant, S1 5′3′ UTR mutant, S1 LM1, S1 LM2 RG) or 0.4 μg of pCI-S2 (for 23/S2/59 RG). Three days post RG, cells were subjected to three freeze thaw cycles. Virus replication was measured by plaque assay on L929 cells and σ protein expression was confirmed by subjecting the RG samples to SDS-PAGE and immunoblotting.

### Figures, graphs and statistical analysis

All graphs and statistical measurements were performed in Graph Pad Prism Version 9 (Graphpad Software). Figures were generated using Adobe Photoshop 2021 and Adobe Illustrator 2021 (Adobe).

### Acknowledgments

We thank previous members of the Miller lab, Dr. Baoqing Guo, Dr. Luke Bussiere, Dr. Nicole Jandick, Dr. Tianjian Tong and Dr. Yijun Qi who have helped us in shaping this project through their valuable suggestions. We thank Jake Peterson from the Moss lab for their valuable insights into understanding the secondary structural predictions of MRV S1 gene, and

Caroline Stokes and Gage Haines, undergraduate students at Iowa State University for their assistance in plasmid design, RG experiments, and immunofluorescence assays. We also thank the Department of Veterinary Medicine and Preventive Medicine, Iowa State University for providing the QuantStudio3 Real-Time PCR system for our RT-qPCR experiments.

## Author Contributions

**Conceptualization:** Debarpan Dhar, Cathy L. Miller.

**Data curation:** Debarpan Dhar, Cathy L. Miller.

**Formal analysis:** Debarpan Dhar, Walter Moss, Cathy L. Miller.

**Funding acquisition:** Cathy L. Miller.

**Investigation:** Debarpan Dhar, Samir Mehanovic, Cathy L. Miller.

**Methodology:** Debarpan Dhar, Cathy L. Miller.

**Resources:** Cathy L. Miller.

**Supervision:** Cathy L. Miller.

**Validation:** Debarpan Dhar.

**Visualization:** Debarpan Dhar.

**Writing – original draft:** Debarpan Dhar, Cathy L. Miller.

**Writing – review & editing:** Debarpan Dhar, Samir Mehanovic, Walter Moss, Cathy L. Miller.

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
