## [Decision Letter · Decision Letter 0]

16 Oct 2023

Dear Dr. Miller,

Thank you very much for submitting your manuscript "Sequences at gene segment termini inclusive of untranslated regions and partial open reading frames play a critical role in mammalian orthoreovirus S gene packaging" for consideration at PLOS Pathogens. As with all papers reviewed by the journal, your manuscript was reviewed by members of the editorial board and by several independent reviewers. In light of the reviews (below this email), we would like to invite the resubmission of a significantly-revised version that takes into account the reviewers' comments.

The reviewers were generally enthusiastic about the work but raised important concerns about the limitations of the reverse genetics assay for documenting viral phenotypes. These are important to address experimentally prior to resubmission.

We cannot make any decision about publication until we have seen the revised manuscript and your response to the reviewers' comments. Your revised manuscript is also likely to be sent to reviewers for further evaluation.

Sincerely,

Anice C. Lowen

Academic Editor

PLOS Pathogens

Alexander Gorbalenya

Section Editor

PLOS Pathogens

Kasturi Haldar

Editor-in-Chief

PLOS Pathogens

orcid.org/0000-0001-5065-158X

Michael Malim

Editor-in-Chief

PLOS Pathogens

orcid.org/0000-0002-7699-2064

The reviewers were generally enthusiastic about the work but raised important concerns about the limitations of the reverse genetics assay for documenting viral phenotypes. These are important to address experimentally prior to resubmission.

Reviewer's Responses to Questions

**Part I - Summary**

Reviewer #1: Many questions remain regarding how multi-segmented dsRNA viruses faithfully package the correct assortment of segments into each virus particle. For ten-segmented reovirus, the 5’ and 3’ termini of RNA segments are predicted to fold into panhandle structures. Previous studies have shown that sequences encompassing the 5’ and 3’ untranslated regions (UTRs) and extending into the ORF are sufficient for segment packaging, including the packaging of foreign sequences. Systems used to make these findings often involved the use of helper viruses or duplicated sequences, which in some cases could confound interpretation of the results. In the current study, Dhar et al. modified the reovirus plasmid-based reverse genetics system to identify RNA sequences required for packaging (inclusive of segment assortment, assembly, and genome replication) through new virus recovery. Their approaches include the use of wobble/block replacement (W/BR) codon usage alterations and plasmid-based ‘transcomplementation’ of viral proteins encoded by altered segments. Using these approaches, they validated previous findings suggesting that the N termini of segments, including 5’ and 3’ untranslated regions (UTRs) and sequences extending into the ORF, are important mediators of reovirus segment packaging. They defined minimal nucleotide sequences sufficient for packaging the S1 segment. Using WB/R and segment incorporation assays, the authors provide evidence that internal segment sequences contribute minimally to RNA packaging. Using RNA structure prediction, the authors identify a six-nucleotide sequence in the 3’ UTR, which is predicted form the loop portion of a stem loop structure, as being conserved across prototype MRV strains. They were unable to recover viable viruses following mutation of these residues, even with transcomplemetation, and conclude that the residues are required for packaging. By engineering segments that encode point mutations in the panhandle that are predicted to disrupt RNA secondary structure or sequence alone, the authors provide evidence that structure rather than sequence is important for segment packaging.

The assay systems presented are new and represent improvements over previous reovirus packaging assays, as they lack the confounders of helper virus or duplicated sequences. One caveat is that, as with previous packaging assay systems, the current system provides little insight into the replication step(s) at which packaging is impaired for viruses that fail to rescue or those that have reduced rescue titers. However, in experiments in which virus is not recovered even in the presence of transcomplemetation, it is now evident that the lack of rescue is not due to a lack of expression of the encoded protein. Some knowledge gained through the study validates prior findings, but several of the current findings are new. Specifically, minimal packaging signals were identified for the reovirus S1 segment, and new insights were made into potential roles of structure versus sequence and a conserved RNA element in the panhandle in packaging. The authors’ conclusions are largely supported by their experimental findings. However, it would inspire even more confidence if conclusions about packaging efficiency were based on more than titers from reverse genetics experiments alone. Although translation of proteins encoded by altered reverse genetic plasmids was compared following transfection of BHK cells, it is possible that they will vary in the presence of the other viral transcripts. Overall, the current work contributes new methods to study viral packaging and new findings made using the assays. These findings make inroads into challenging questions related to the mysteries of RNA segment assortment and new virus assembly that have long challenged the dsRNA virus field.

Reviewer #2: Using mammalian orthoreovirus (MLV), this manuscript by Dhar et al describes a clever approach for defining 5’ and 3’ terminal regions requiring nucleotide conservation for efficient virus rescue and replication. Specifically, they introduced wobble mutations into increase intergenic regions and monitored for loss of virus rescue and plaque size. The authors furthermore excluded that their modifications were merely reducing the levels of viral mRNAs and proteins expressed from the reverse genetics plasmids. While I cannot be entirely sure that the identified 3’ and 5’ sequences are all specifically essential for “packaging” as stated (explained further below), this study clearly defines the extremities of MLV that can tolerate nucleotide modifications. The researchers then applied their findings to incorporate a fully exogenous gene-encoding segment into progeny MLVs, albeit these progenies were replication defective as would be expected. Overall, the approach is clever and the experiments comprehensively done and described. The findings help define the nucleotide-sequence-dependent regions of MRV at 3’and 5’ ends, but moreover will appeal to a more general audience for the approach.

Minor suggested changes

- I would suggest providing alternative interpretations to absence of infectious virus rescue that are still compatible with the results but not necessarily a direct reflection of packaging. As an example, if progeny virus particle packaged a genome but required nucleotide sequence or structure for onset of negative sense RNA synthesis etc. While assembly is most likely, I am not convinced it is the only RNA-dependent process between mRNA/protein expression from the reverse genetics plasmids (measured) to viruses capable of producing titer (measured).

- Line 104 – the statement that the specific infectivity of reovirus is <2 (particle/PFU) uses references that are older than many more-recent assessments that show a ratio of ~100-300 on average. As one example, see PMID: 22811534; but many studies find similar ranges. This still does not affect the major point of your argument, since random reassortment of 10 segments would give a much lower specific infectivity than 100-300.

- For references in lines 110-118, it would help to indicate the virus these steps were discovered with as they may/may not directly apply to MLV and as written, it rather seems as if the references are providing proof that it is for MLV specifically.

- In lines 133-147 describing previous work on identifying packaging signals, it seems important to describe what the packaging signals were discovered to me, and why there was a need to further examine this question.

Reviewer #3: Dhar et al

In this paper, Dhar et al define determinants in the mammalian reovirus S segments that control packaging efficiency. Using a clever strategy, the authors generate mutant viral gene segments such that the entire ORF is wobbled as much as possible. This strategy ensures that putative packaging signals are disrupted without changes in the amino acid sequence. Using this strategy, the authors systematically analyze the sequences in the viral genome segments necessary to allow packaging of the viral gene segments. With particular focus on the viral S1 gene segment, the authors identify the minimal sequence necessary. These sequences extend beyond the UTR regions. They also uncover a role for a previously unidentified panhandle structure and loop region in the UTR for packaging. Finally, they identify regions that are sufficient to allow packaging of a gene segment containing a trans gene using a complementation system

The experiments in this paper are clear, well presented and carefully controlled. I particularly appreciate the care taken to ensure that mutations introduced by the authors do not influence the other functions of the genome segment such as RNA synthesis and translation. The findings are new and exciting and will make an important new contribution to the field.

The paper can be improved in the following ways

1. Packaging is being measured by surrogate method - ie recovery of virus. It keeps open the possibility that packaging does indeed occur but the virus produced is non-infectious. The authors should consider alternate methods - such as qPCR from “virus fraction” to assess packaging. Perhaps showing that with one a few select mutants will be helpful.

2. In the original paper on reovirus reverse genetics (Kobayashi et al 2007), it was shown that after transfection of T7 vectors expressing reovirus RNAs, very few cells are capable of producing virus. In other words, only a few received all 10 plasmids. Recovery of sufficient virus titer after reverse genetics required incubation over several days and the spread of virus to new cells that did not receive all 10 plasmids. Therefore, the experiments showing western blots for sigma1 expression are a little bit puzzling. One would expect very low level detection (if any) for viruses that are not recoverable. Yet, the expression seems reasonably high. It might be helpful to the reader if the authors demonstrate that this expression does not require spread of virus through the population - perhaps by blocking secondary rounds of infection through addition of entry inhibitors or neutralizing antisera.

**Part II – Major Issues: Key Experiments Required for Acceptance**

Reviewer #1: The consistency of rescue titers from experiment to experiment, combined with measures of protein expression and transcript levels from reverse genetics plasmids separately transfected in BHK cells, provides confidence that reverse genetics recovery titer is correlated with packaging efficiency rather than just indicating that viable virus recovery is possible. However, whether these factors are directly related is not entirely clear, as other factors could influence the outcome of rescue experiments. Quantitation of replication kinetics for recovered viruses with altered segments relative to those of the parental viruses, at least for those viruses that do not require transcomplementation, would provide an important comparison. This comparison would involve multiple rounds of amplification. It is likely that the same is true following a five-day reverse genetics transfection. However, in the case of a replication assay, an equal infectious dose would be used as the experimental input. In the case of a reverse genetics experiment, a substantial delay in initial virus recovery for any number of reasons might be amplified.

Protein expression levels in BHK cells for altered segment constructs all were measured in the absence of expression of other viral transcripts/proteins. It is possible that expression levels differ for altered transcripts during reverse genetics experiments when all other viral transcripts are present in the cell and available to compete. It appears that much larger amounts of plasmid (10 µg) were transfected for protein expression experiments than for reverse genetics experiments (0.4 µg). If it is it possible to co-transfect the other RG plasmids and still detect expression of the protein of interest (e.g., 1-2 µg of each plasmid), then these experiments should be done to determine whether the presence of the other viral transcripts/proteins affects protein levels for altered constructs.

Reviewer #2: NA

Reviewer #3: 1. A direct measurement of packaging for select set of mutants would be helpful to ensure that its truly a packaging defect.

2. Experimental proof or an explanation of how sufficient protein is produced for nonrecoverable viruses would be helpful.

**Part III – Minor Issues: Editorial and Data Presentation Modifications**

Reviewer #1: Line 396-407: It is unclear why different RNA structure prediction algorithms were used to predict structures for S1 gene sequences from the three major MRV serotypes or a single S1 segment. Details of alignment strategies are not described in Materials and Methods. Was any setting other than the default applied?

Line 488-490: RNA secondary structure and its disruption through engineered mutations still are only predicted computationally, though the current findings do provide additional support for the predictions. Perhaps phrasing could be altered slightly to indicate that the structures still are predictions.

Line 624-639: While general construction of wobble mutated S gene segment-encoding plasmids is described, their sequences are not reported or deposited. Thus, it is not possible for another group to replicate the study.

Line 711-723: Some details of the segment incorporation assay in which S1 terminal sequences were tested for the potential to mediate packaging of NanoLuc are unclear. Were equal volumes of lysate from each experiment used to infect the next round of cells (most likely, yes)? Please describe the control treatment with pNL3.1 to which infections were compared in Fig. 6B. Was this simply a plasmid transfection conducted alongside the infection? Also, why were DNase I treated samples co-infected with WT T1L rather than just being added to L929 cells? Please very briefly describe the purpose of the co-infecting virus.

Reviewer #2: Minor suggested changes

- I would suggest providing alternative interpretations to absence of infectious virus rescue that are still compatible with the results but not necessarily a direct reflection of packaging. As an example, if progeny virus particle packaged a genome but required nucleotide sequence or structure for onset of negative sense RNA synthesis etc. While assembly is most likely, I am not convinced it is the only RNA-dependent process between mRNA/protein expression from the reverse genetics plasmids (measured) to viruses capable of producing titer (measured).

- Line 104 – the statement that the specific infectivity of reovirus is <2 (particle/PFU) uses references that are older than many more-recent assessments that show a ratio of ~100-300 on average. As one example, see PMID: 22811534; but many studies find similar ranges. This still does not affect the major point of your argument, since random reassortment of 10 segments would give a much lower specific infectivity than 100-300.

- For references in lines 110-118, it would help to indicate the virus these steps were discovered with as they may/may not directly apply to MLV and as written, it rather seems as if the references are providing proof that it is for MLV specifically.

- In lines 133-147 describing previous work on identifying packaging signals, it seems important to describe what the packaging signals were discovered to me, and why there was a need to further examine this question.

Reviewer #3: (No Response)

PLOS authors have the option to publish the peer review history of their article (what does this mean?). If published, this will include your full peer review and any attached files.

Reviewer #1: No

Reviewer #2: No

Reviewer #3: No
---

## [Decision Letter · Decision Letter 1]

8 Feb 2024

Dear Dr. Miller,

We are pleased to inform you that your manuscript 'Sequences at gene segment termini inclusive of untranslated regions and partial open reading frames play a critical role in mammalian orthoreovirus S gene packaging' has been provisionally accepted for publication in PLOS Pathogens.

Best regards,

Anice C. Lowen

Academic Editor

PLOS Pathogens

Alexander Gorbalenya

Section Editor

PLOS Pathogens

Michael Malim

Editor-in-Chief

PLOS Pathogens

orcid.org/0000-0002-7699-2064

Editors:

Thank you for your thoughtful handling of the prior critiques. The reviewers indicated that the revisions have brought important improvements and that the work substantively advances the field.

Reviewer Comments (if any, and for reference):

Reviewer's Responses to Questions

**Part I - Summary**

Reviewer #1: Many questions remain regarding how multi-segmented dsRNA viruses faithfully package the correct assortment of segments into each virus particle. For ten-segmented reovirus, the 5’ and 3’ termini of RNA segments are predicted to fold into panhandle structures. Previous studies have shown that sequences encompassing the 5’ and 3’ untranslated regions (UTRs) and extending into the ORF are sufficient for segment packaging, including the packaging of foreign sequences. Systems used to make these findings often involved the use of helper viruses or duplicated sequences, which in some cases could confound interpretation of the results. In the current study, Dhar et al. modified the reovirus plasmid-based reverse genetics system to identify RNA sequences required for packaging (inclusive of segment assortment, assembly, and genome replication) through new virus recovery. Their approaches include the use of wobble/block replacement (W/BR) codon usage alterations and plasmid-based ‘transcomplementation’ of viral proteins encoded by altered segments. Using these approaches, they validated previous findings suggesting that the N termini of segments, including 5’ and 3’ untranslated regions (UTRs) and sequences extending into the ORF, are important mediators of reovirus segment packaging. They defined minimal nucleotide sequences sufficient for packaging the S1 segment. Using WB/R and segment incorporation assays, the authors provide evidence that internal segment sequences contribute minimally to RNA packaging. Using RNA structure prediction, the authors identify a six-nucleotide sequence in the 3’ UTR, which is predicted form the loop portion of a stem loop structure, as being conserved across prototype MRV strains. They were unable to recover viable viruses following mutation of these residues, even with transcomplemetation, and conclude that the residues are required for packaging. By engineering segments that encode point mutations in the panhandle that are predicted to disrupt RNA secondary structure or sequence alone, the authors provide evidence that structure rather than sequence is important for segment packaging.

The assay systems presented are new and represent improvements over previous reovirus packaging assays, as they lack the confounders of helper virus or duplicated sequences. Some knowledge gained through the study validates prior findings, but several of the current findings are new. Specifically, minimal packaging signals were identified for the reovirus S1 segment, and new insights were made into potential roles of structure versus sequence and a conserved RNA element in the panhandle in packaging. In the initial submission, a concern was that the system provided little insight into the replication step(s) at which packaging is impaired for viruses with reduced rescue titers. In the revised manuscript, experiments that used identical input virus titers for S1 WT, 200/S1/200, and 18/S1/40 and quantified infected cells or virus titer following multiple rounds of replication point to a late replication step, after entry, transcription, and translation. The authors’ conclusions are largely supported by their experimental findings. A concern that translation of proteins encoded by altered reverse genetic plasmids may vary in the presence of the other viral transcripts has been reasonably addressed in the revised manuscript. Overall, the current work contributes new methods to study viral packaging and new findings made using the assays. These findings make inroads into challenging questions related to the mysteries of RNA segment assortment and new virus assembly that have long challenged the dsRNA virus field.

Reviewer #3: This is revised manuscript from the group of Dr. Cathy Miller. In this work, the authors perform experiments to identify mechanisms by which genome segments are packaged. They identify areas where packaging signals are present and highlight a role of a panhandle structure and specific sequences therein.

**Part II – Major Issues: Key Experiments Required for Acceptance**

Reviewer #1: (No Response)

Reviewer #3: In my previous review, I raised several concerns. Some similar concerns were also raised by the other reviewers. The authors provide some new experimental evidence to support their conclusions and clarify some experiments they performed. They also explain why additional concerns cannot be easily addressed. I am satisfied by their response.

**Part III – Minor Issues: Editorial and Data Presentation Modifications**

Reviewer #1: Line 261-263. Although the protein expression patterns for cells in which a single plasmid or all RG plasmids were transfected appear similar, one would need to include all samples on the same immunoblot to directly compare the two conditions.

Sequences for “wobble” plasmids are reported to be available through ISU DataShare, whose identifier Is noted in the Data Availability Statement. Many support files were located, but the sequences were not easily located, though I trust that they will be accessible by the time of publication.

All other minor comments were sufficiently addressed in the revised manuscript.

Reviewer #3: (No Response)

PLOS authors have the option to publish the peer review history of their article (what does this mean?). If published, this will include your full peer review and any attached files.

Reviewer #1: No

Reviewer #3: No

---

## [Editor Report · Acceptance letter]

19 Feb 2024

Dear Dr. Miller,

We are delighted to inform you that your manuscript, "Sequences at gene segment termini inclusive of untranslated regions and partial open reading frames play a critical role in mammalian orthoreovirus S gene packaging," has been formally accepted for publication in PLOS Pathogens.

Best regards,

Michael Malim

Editor-in-Chief

PLOS Pathogens

orcid.org/0000-0002-7699-2064